# Helical supramolecular polymers with rationally designed binding sites for chiral guest recognition

Krishnachary Salikolimi[1], Vakayil K. Praveen [2], Achalkumar Ammathnadu Sudhakar [3✉], Kuniyo Yamada[1], Noriko Nishizawa Horimoto [1] & Yasuhiro Ishida [1✉]

Since various helical supramolecular polymers became available, their application to molecular chirality recognition have been anticipated but not extensively studied. So far, only a few examples of chiral reactions have been reported, but none for chiral separation. Here, we report the application of a helical supramolecular polymer to the enantio-separation of chiral guest molecules. The monomer of this supramolecular polymer is the salt-pair of a dendritic carboxylic acid with an enantiopure amino alcohol. In an apolar solvent, this salt-pair stacks via hydrogen bonds to form a helical polymer. In conjunction with this carboxylic acid, various amino alcohols afford supramolecular polymers, whose helical handedness is determined by the stereochemistry of the amino alcohols. When two salts with the same chirality are mixed, they undergo copolymerization, while those with opposite chirality do not. Owing to this stereoselective copolymerizability, the helical supramolecular polymer could bias the enantiomeric composition of chiral amino alcohols.

[1] RIKEN Center for Emergent Matter Science, 2-1 Hirosawa, Wako, Saitama 351-0198, Japan. [2] Photosciences and Photonics Section, Chemical Sciences and Technology Division, CSIR-National Institute for Interdisciplinary Science and Technology (CSIR-NIIST), Thiruvananthapuram, Kerala 695 019, India. [3] Department of Chemistry, Indian Institute of Technology Guwahati, Guwahati, Assam 781 039, India. ✉email: achalkumar@iitg.ac.in; y-ishida@riken.jp

Chirality transfer between small molecules and polymers is one of the most seminal concepts in the field of molecular chirality sciences (Fig. 1a), because these chemical classes take roles complementary to each other. For small molecules, various enantiopure species are readily available, such as amino acids and saccharides, while polymers tend to adopt stable helical conformation that is suitable for the storage, transfer, and amplification of chiral information. The history of the concept "molecule–polymer chirality transfer" started from the pioneering discoveries that chiral small molecular units can induce or bias the conformational helicity of polymers through covalent[1,2] and non-covalent[3,4] interactions (Fig. 1a, i). An equally important finding is that helical polymers can provide superb chiral environment, reminiscent of the guest-binding sites of enzymes, for the separation and creation of chiral small molecules, as represented by stationary phases for chiral chromatography[5–7] and polymer catalysts for asymmetric synthesis (Fig. 1a, ii)[8].

Later, the scope of polymers was expanded to those with non-covalently formed backbones, the so-called "non-covalent polymers" or "supramolecular polymers", which opened up the new possibilities for polymer sciences[9–17]. Similar to covalent polymers, one-handed helicity of supramolecular polymers can be induced by the covalent[18] and non-covalent[14,19–21] interactions of chiral small molecular units (Fig. 1a, iii). Considering the analogy of the schemes in Fig. 1, chirality transfer from "supramolecular polymers" to "small molecules" should be possible (Fig. 1a, iv), but has not been extensively studied. So far, several examples of asymmetric catalysis have been reported[22–27], while the reports on enantio-separation have been rare, which would need further elaborated guest-binding sites. To realize this chirality transfer, one serious problem is the difficulty in designing non-covalent interaction sites in supramolecular polymers, which should work simultaneously for two purposes, i.e., polymer-backbone formation and guest-molecule recognition. For each of these purposes, multiple non-covalent interactions, such as electrostatic, dipole–dipole, metal–ligand, hydrogen-bonding, π–π, and solvophobic interactions, should work cooperatively and directionally. However, such interactions often interfere with each other, which reduces the efficiency of polymerization and chirality recognition.

Here, we report the application of a helical supramolecular polymer to the enantio-separation of chiral guest molecules. Our strategy is very simple, to use the non-covalent interaction sites in the supramolecular polymer not only for the connection of monomers, but also for the recognition of chiral guest molecules. This idea has not been extensively studied but seems reasonable, because some supramolecular polymers are formed via homo-chiral polymerization[28–32], where the elongating polymers selectively accommodate the monomers with the same chirality as their constituents. To use this stereoselective process for the enantio-separation of chiral molecules, we designed a family of helical supramolecular polymers, whose monomer is composed of an achiral main body (dendritic carboxylic acid 1; Fig. 1b, upper)[33] and a chiral auxiliary (amino alcohol $X = 2^S/2^R$, $3^S/3^R$, $4^S/4^R$…; Fig. 1b, middle), which electrostatically interact to form a salt-pair (1·X). When dissolved in dodecane, the salt-pair 1·X stacks via hydrogen bonds to form a helical polymer. As amino alcohols are a typical "chiral pool" that can offer various species in enantiopure form for both antipodes[34], a small library of helical supramolecular polymers can be easily prepared without tough synthesis (Fig. 1b). The helical handedness of these polymers is determined by the stereochemistry of the amino alcohols; those with S and R configurations afford right- and left-handed helices, respectively. When two kinds of salts with the same chirality (e.g., 1·2$^S$ and 1·3$^S$) are mixed in solution, they undergo copolymerization (Fig. 1c, left), while those with the opposite

chirality (e.g., 1·2$^S$ and 1·3$^R$) do not (Fig. 1c, middle). Thus, the polymer of 1·2$^S$ can recognize the chirality of the guests 3$^S$/3$^R$, where the homo-chiral guest 3$^S$ is accommodated, while the hetero-chiral guest 3$^R$ is rejected (Fig. 1c, right). Owing to this stereoselective copolymerizability, the polymer of 1·2$^S$ can bias the enantiomeric ratio of 3$^S$ and 3$^R$ through a simple one-pot process.

## Results

**Preparation and characterization of the helical supramolecular polymers**. As the design of our supramolecular polymers is uncommon, we started by studying their fundamental properties, choosing the polymer composed of 1 (Supplementary Methods and Supplementary Fig. 1)[34] and 2$^S$ as a representative example (Fig. 2). In dodecane at 298 K, 2$^S$ was hardly soluble by itself (Fig. 2a, left), while the salt 1·2$^S$, prepared by dissolution of equimolar 1 and 2$^S$ in CHCl$_3$ and subsequent evaporation of CHCl$_3$ (Supplementary Methods), was readily soluble in dodecane (Fig. 2a, right). Fourier transform infrared absorption (FT-IR) spectroscopy revealed that the molecules of 1 and 2$^S$ in dodecane (4.0 mM) existed as the salt-pairs of deprotonated 1 (R-CO$_2^-$) and protonated 2$^S$ (R'-NH$_3^+$), as evidenced by the absorption of the carboxylate (R-CO$_2^-$) at 1575 cm$^{-1}$, together with the lack of the absorption of the free carboxylic acid (R-CO$_2$H) that should appear at 1700 and 1750 cm$^{-1}$ (Supplementary Methods and Fig. 2b).

Dynamic light scattering (DLS) analysis proved that 1·2$^S$ in dodecane (4.0 mM) formed polymeric aggregates with hydrodynamic diameter of 20 nm (Supplementary Fig. 2a, upper, 298 K). This polymerization was promoted by the hydrogen-bonding interactions between the salt-pairs 1·2$^S$; such polymeric aggregates did not form when the hydrogen-bonding sites in 2$^S$ were protected by methylation (Supplementary Fig. 3a–c). Owing to the dynamic nature of hydrogen bonds, the DLS hydrodynamic diameter of the polymeric aggregates of 1·2$^S$ was decreased from 20 to 1 nm as temperature increased from 298 to 348 K (Supplementary Fig. 2a). Atomic force microscopy (AFM) indicated that the polymeric aggregates were one-dimensional stacks of the salt-pair 1·2$^S$ with >200 nm length and ~4 nm width (Fig. 2f).

In dodecane (4.0 mM) at 298 K, 1·2$^S$ exhibited strong circular dichroism (CD) signals at the absorption region of 1 (Supplementary Methods and Supplementary Fig. 4), whose CD spectrum was a perfect mirror image of its antipode 1·2$^R$ (Fig. 2c). Emergence of these strong CD signals was accompanied by the formation of the supramolecular polymer, suggesting its helical conformation. Indeed, the CD signals vanished when the polymerization of 1·2$^S$ was suppressed by heating to 348 K (Supplementary Fig. 2b) or by protecting the hydrogen-bonding sites in 2$^S$ (Supplementary Fig. 3d–f). The CD signals were useful for gaining further insights into this supramolecular polymer. First, the CD-monitored titration experiment at 298 K proved that this helical polymeric architecture was formed with a 1:1 stoichiometry of 1 and 2$^S$ (Fig. 2d and Supplementary Fig. 5). In addition, the CD-monitored dilution experiment at 298 K showed that this helical polymeric architecture dissociated when monomer concentration was decreased to less than 0.5 mM (Supplementary Fig. 6). Therefore, all of the following ultraviolet absorption (UV) and CD measurements were conducted at a high monomer concentration (4.0 mM) using an optical cuvette with a short pathlength (0.1 mm) for appropriate optical density.

By cooling the dodecane solution of 1·2$^S$ from 348 to 283 K and monitoring the CD intensity at 354 nm (Supplementary Methods and Supplementary Fig. 4), a cooling curve was obtained (Fig. 2e, blue), whose less sigmoidal shape suggests that the

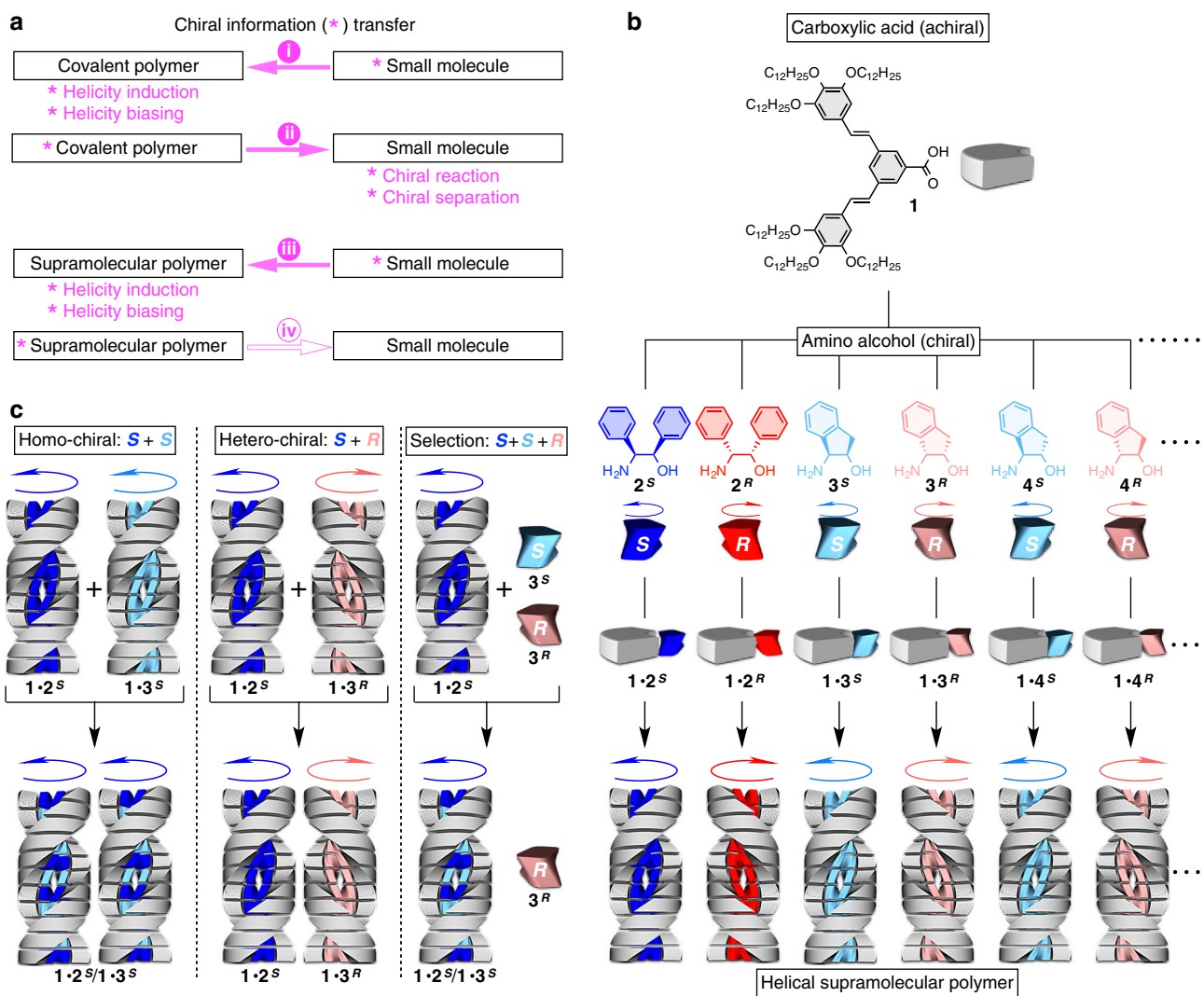

**Fig. 1 Design of helical supramolecular polymers with modulable chiral auxiliaries and their application to homo/hetero-chiral recognition. a** Schemes for chirality transfer between covalent polymers and small molecules [(i) and (ii)] and between supramolecular polymers and small molecules [(iii) and (iv)]. **b** Design of the helical supramolecular polymers in this work, whose monomer consists of an achiral main body (dendritic carboxylic acid **1**) and a chiral auxiliary (amino alcohol **X** = **2$^S$/2$^R$**, **3$^S$/3$^R$**, **4$^S$/4$^R$**...) that electrostatically associate to form a salt-pair (**1·X**). The monomer **1·X** stacks via hydrogen bonds to form the supramolecular polymer. **c** Stereoselective copolymerization of two kinds of salts. When mixed in solution, two kinds of salts with the same chirality (**1·2$^S$** and **1·3$^S$**) are copolymerizable (left), while those with opposite chirality (**1·2$^S$** and **1·3$^R$**) are not (middle), because of the matching and mismatching of their helical-handedness preference. Owing to this stereoselective copolymerizability, the polymer of **1·2$^S$** accommodate homo-chiral **3$^S$** and reject hetero-chiral **3$^R$** (right).

polymerization proceeded via not an isodesmic but a cooperative mechanism (Fig. 2g, ii–iv) with an elongation temperature ($T_e$) of 335 K[18]. We also confirmed that an essentially identical cooling curve was obtained by monitoring the UV absorption at 385 nm (Fig. 2e, light blue). In the following part, we used CD-monitored cooling curves to discuss the polymer elongation profiles, because our supramolecular polymers showed very small changes in UV absorption (Supplementary Fig. 2c, lower). Unlike the CD and UV spectra, the IR spectrum showed sluggish temperature-dependent changes, where a large fraction of the carbonyl group in **1** remained in the carboxylate (R-CO$_2^-$) form even at 348 K (Supplementary Fig. 2d). This observation is consistent with the fact that salt-pair interactions between ammoniums and carboxylates involve electrostatic attraction, and therefore are generally stronger than normal hydrogen-bonding interactions[35]. Thus, the salt unpairing (Fig. 2g, i) occurred in a temperature region higher than that of polymer elongation ($T_e$ = 335 K).

In crystalline states, it is known that the salts of carboxylic acids with amino alcohols generally form a hydrogen-bonded helical columnar network as depicted in Fig. 2h, in which the carboxylic acid (R-CO$_2^-$ form) provides four hydrogen-accepting sites, while the amino alcohol (R'-NH$_3^+$ form) provides four hydrogen-donating sites (Supplementary Fig. 7a)[36,37]. According to our survey of the Cambridge Structural Database (CSD), 19 kinds of carboxylate salts of **2$^S$** (**2$^R$**) adopt hydrogen-bonded networks similar to Fig. 2h (Supplementary Fig. 7b, c). Considering the crucial role of hydrogen-bonding interactions in the supramolecular polymerization of **1·2$^S$** (Supplementary Fig. 3a–c), as well as the width of the resultant polymeric fibers observed in AFM (3.8 nm; Supplementary Fig. 11a), hydrogen-bonded network similar to Fig. 2h would also form in the supramolecular polymer of **1·2$^S$**. In relation to this, we recently reported that the salt of a carboxylic acid with a chiral amino alcohol assembled into a double helix structure in its liquid

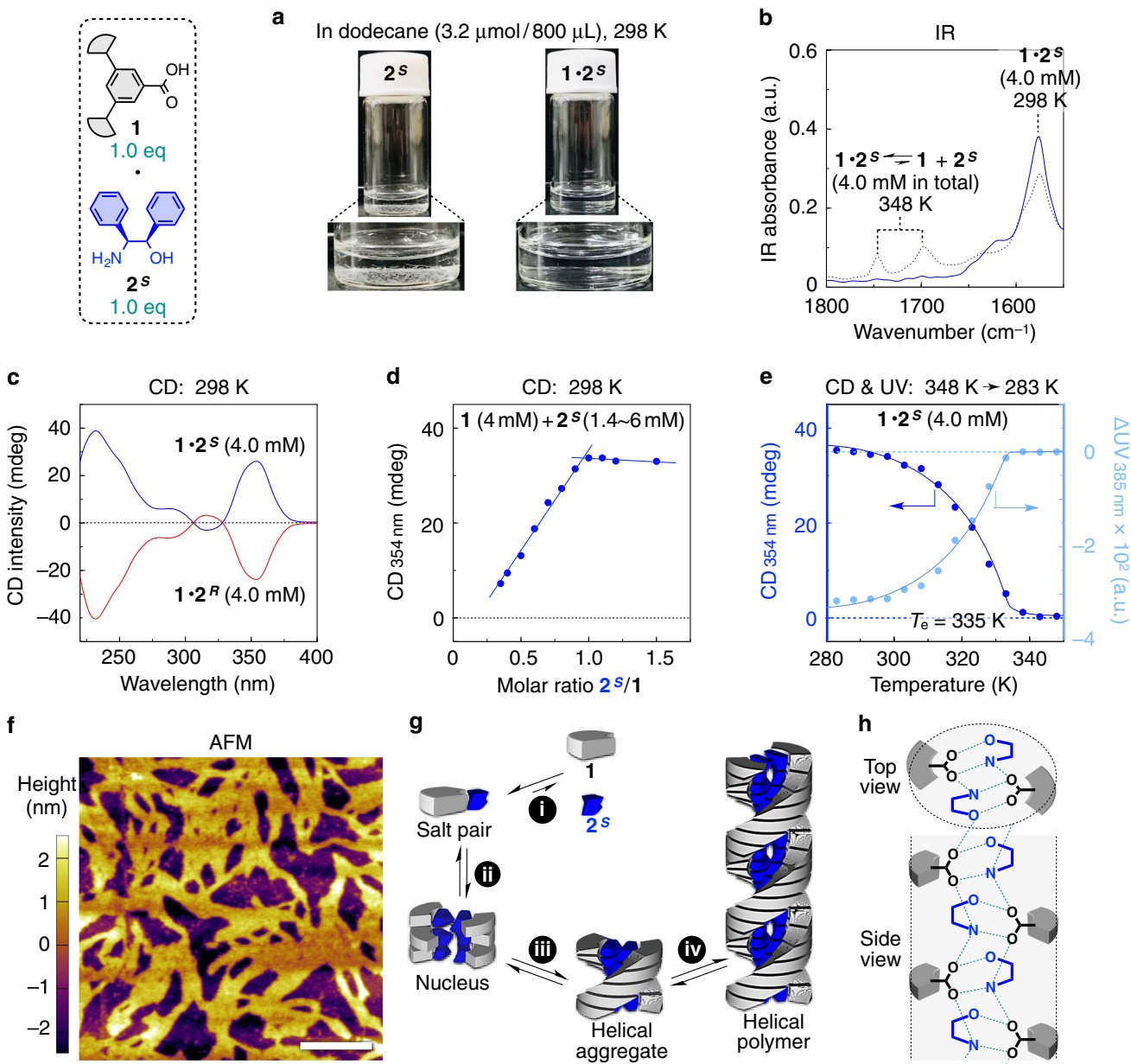

**Fig. 2 Preparation and characterization the helical supramolecular polymer. a** Solubility of **2**$^S$ (left) and their salt **1·2**$^S$ (right) in dodecane (3.2 μmol/800 μL) at 298 K. **b** FT-IR spectra of **1·2**$^S$ in dodecane (4.0 mM) at 298 (solid curve) and 348 K (dashed curve). **c** CD spectra of **1·2**$^S$ and **1·2**$^R$ in dodecane (4.0 mM) at 298 K. **d** CD (354 nm)-monitored titration of **1** (4.0 mM) with **2**$^S$ (1.4–6.0 mM) in dodecane at 298 K. **e** Cooling curves from 348 to 283 K at –1.0 K/min of **1·2**$^S$ in dodecane (4.0 mM) monitored with CD intensity at 354 nm and UV absorption at 385 nm. **f** AFM image of **1·2**$^S$ spin-coated on a HOPG substrate from a dodecane solution (4.0 mM). Scale bar, 200 nm. For a magnified image, see Supplementary Fig. 10a. **g** Schematic for the supramolecular polymerization of **1** and **2**$^S$. **h** Helical hydrogen-bonded network generally formed in the crystal structures of carboxylate salts of **2**$^S$ (**2**$^R$). See also Supplementary Fig. 7.

crystalline state, where the precise positions of the molecular components were obtained based on the determination of space group ($P6_122$) by X-ray diffraction (XRD) studies[38]. In this work, **1·2**$^S$ was found to exhibit a thermotropic liquid crystalline phase, whose XRD pattern (Supplementary Methods) was quite resembling to that in our previous report[38]; all reflections were elucidated on the supposition of the same space group ($P6_122$; for details, see Supplementary Fig. 8). Therefore, a double helix structure similar to that we reported previously[38] was considered to be a possible structural model of the supramolecular polymer of **1·2**$^S$.

Not only **2**$^S$ (**2**$^R$) but also various chiral amino alcohols could form supramolecular polymers by dissolving their salts with **1** in

dodecane (Supplementary Fig. 9). The CD intensity of these supramolecular polymers significantly depended on the amino alcohol units. The supramolecular polymers prepared from amino alcohols **2**$^S$–**6**$^S$ (**2**$^R$–**6**$^R$) having two substituents at both of the C$^1$ and C$^2$ positions, including those with a fused-ring structure **3**$^S$ (**3**$^R$) and **4**$^S$ (**4**$^R$), generally exhibited large CD signals at the absorption region of **1** (Supplementary Fig. 9a–e). Meanwhile, amino alcohols **7**$^S$ (**7**$^R$) and **8**$^S$ (**8**$^R$) having one substituent at either of the C$^1$ and C$^2$ positions afforded supramolecular polymers with small CD signals (Supplementary Fig. 9f, g). To form a helical supramolecular polymer, the conformational fixation of the amino alcohol unit by the interference of the two substituents is crucial[39]. Among the family of supramolecular

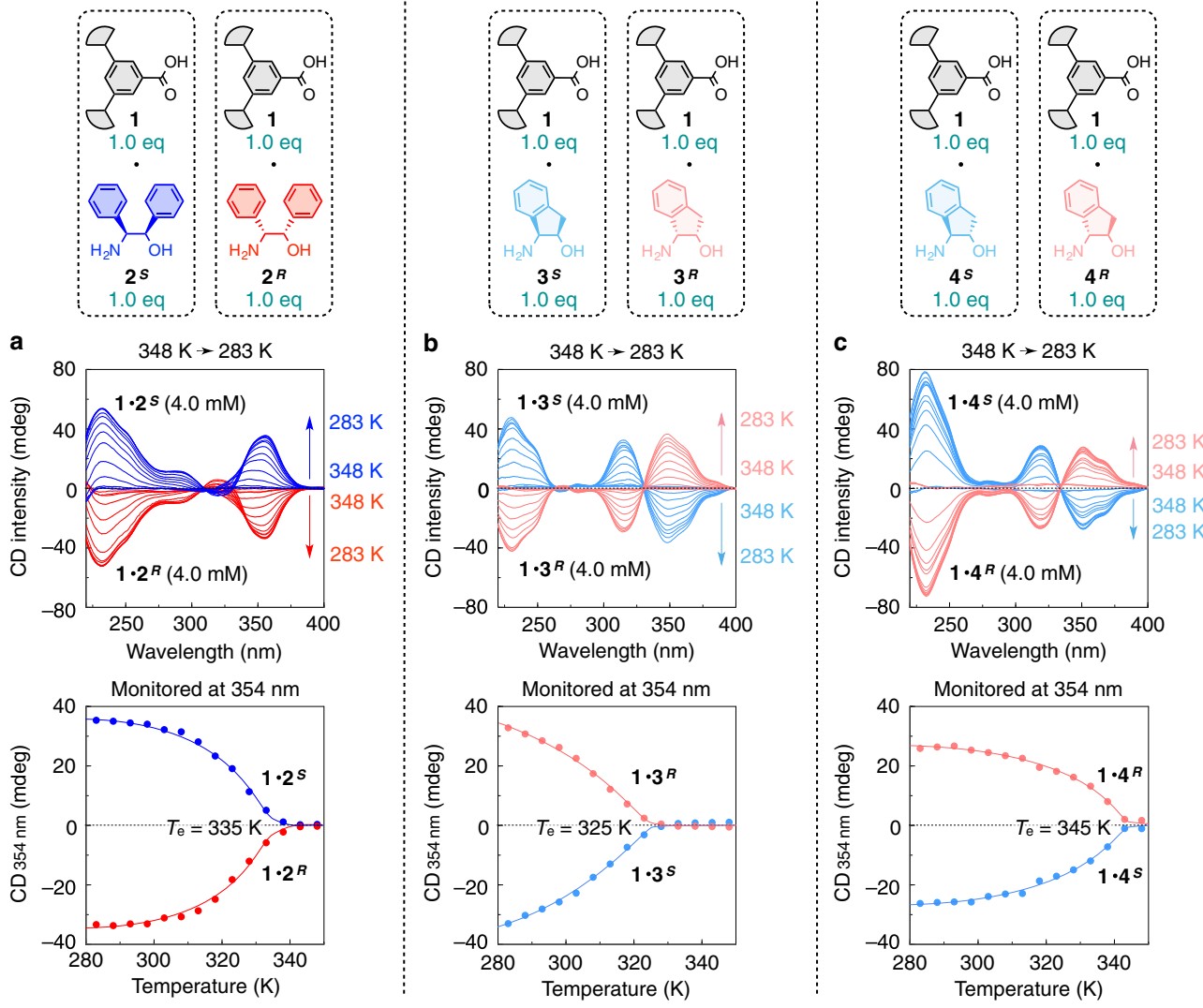

**Fig. 3 Elongation profiles of the helical supramolecular polymers upon cooling. a–c** Variable-temperature CD profiles at every 5 K step of **1·2$^S$/1·2$^R$** (**a**), **1·3$^S$/1·3$^R$** (**b**), and **1·4$^S$/1·4$^R$** (**c**) in dodecane (4.0 mM) measured upon cooling from 348 to 283 K at –1.0 K/min. CD spectra (upper) and cooling curves monitored with CD intensity at 354 nm (lower).

polymers obtained (Supplementary Fig. 9), those of **1·2$^S$**, **1·3$^S$**, and **1·4$^S$** (**1·2$^R$**, **1·3$^R$**, and **1·4$^R$**) showed relatively large CD signals, and therefore were chosen for further studies. According to their CD-monitored cooling curves with less sigmoidal shapes, all of these salts are also likely to polymerize via a cooperative mechanism, where the elongation temperatures ($T_e$) of **1·2$^S$**, **1·3$^S$**, and **1·4$^S$** (**1·2$^R$**, **1·3$^R$**, and **1·4$^R$**) were estimated to be 335, 325, and 345 K, respectively (Fig. 3a–c).

The $T_e$ values of the supramolecular polymers were highly dependent on the amino alcohol units (**4$^S$** > **2$^S$** > **3$^S$**). Hypothesizing that this order was in relation to the conformational stability of the amino alcohols, we retrieved all crystal structures of the salts of **2$^S$**, **3$^S$**, and **4$^S$** (including their derivatives and their antipodes) that were deposited in the Cambridge Structural Database (CSD), and checked the conformation of the amino alcohols, focusing on the following two points: (i) rotation around the C$^1$–C$^2$ bond (Supplementary Fig. 10a–c) and (ii) orientation of the aromatic substituent(s) (Supplementary Fig. 10d–f)[39]. Concerning the point (i), **2$^S$** and **4$^S$** showed a strong preference only to a single state (Supplementary Fig. 10a, c), while **3$^S$** adopted two states (Supplementary Fig. 10b). The exceptionally low $T_e$ of the supramolecular polymer of **1·3$^S$** would be due to the

C$^1$–C$^2$ bond rotation, which changes the direction of the hydrogen-bonding sites in **3$^S$**. Such a clear difference between **3$^S$** and **4$^S$**, despite their apparently similar structure, can be elucidated by the number of axial/equatorial substituents in their cyclopentane cores. In principle, **3$^S$** and **4$^S$** can possibly adopt two states in terms of point (i). In the case of **4$^S$**, one state has no axial substituent, while the other state has two axial substituents (Supplementary Fig. 10c), and consequently, the equilibrium is highly biased to the former state. Contrarily, in the case of **3$^S$**, each of the two states have one axial substituent (Supplementary Fig. 10b), so that **3$^S$** undergoes conformational changes between the two states. On the other hand, concerning the point (ii), the orientation of the aromatic group in **4$^S$** was fixed, probably due to its fused-ring structure (Supplementary Fig. 10f). Meanwhile the two aromatic groups in **2$^S$** that lacks the fused-ring structure rotated into various directions (Supplementary Fig. 10d). Taking account of the points (i) and (ii), the order of conformational stability was considered to be **4$^S$** > **2$^S$** > **3$^S$**, which was consistent with the order of $T_e$.

The helical handedness of these supramolecular polymers could be determined by AFM[40], where the polymers were deposited onto highly oriented pyrolytic graphite (HOPG) substrates from their

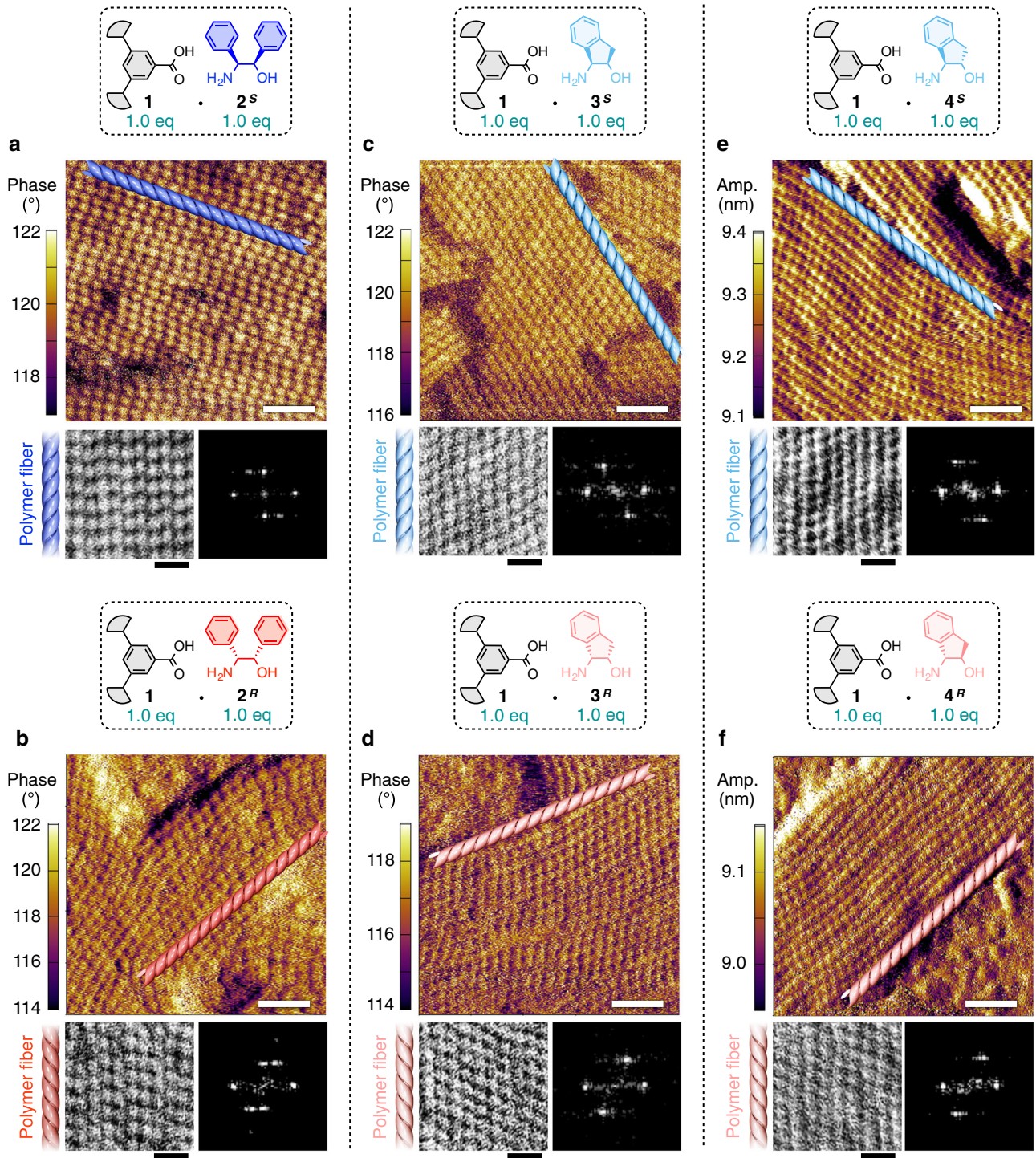

**Fig. 4 Determination of the helical handedness of the supramolecular polymers. a–f** AFM data of **1•2$^S$** (**a**), **1•2$^R$** (**b**), **1•3$^S$** (**c**), **1•3$^R$** (**d**), **1•4$^S$** (**e**), and **1•4$^R$** (**f**) spin-coated on HOPG substrates from dodecane solutions (4.0 mM). AFM phase/amplitude images (upper; scale bar, 20 nm), gray-scale images trimmed (38 × 38 nm) from the AFM images (lower left; scale bar, 10 nm), and magnified FFT images of the gray-scale images (lower right). The illustrations of helices indicate the orientation direction of the supramolecular polymer fibers. The corresponding AFM height images are shown in Supplementary Fig. 11. For details of FFT, see Supplementary Methods and Supplementary Fig. 12.

dodecane solutions (Supplementary Methods, Fig. 4, and Supplementary Fig. 11). Their AFM images (100 × 100 nm) showed a lot of parallelly packed fibers, which consisted of ellipsoidal segments stacked along the fiber axes, corresponding to the helical pitches (Fig. 4a–f, upper). The helical handedness could be determined by the inclination direction of the ellipses with respect to the fiber axis (clockwise [right-handed] and counter clockwise [left-handed]).

For further validation, a square region (38 × 38 nm) was trimmed from each AFM image (Fig. 4a–f, lower left) and subjected to fast Fourier transform (FFT) to obtain an image (Fig. 4a–f, lower right) that showed dissymmetrically arranged spots representing the helical handedness (Supplementary Methods and Supplementary Fig. 12)[40]. Interestingly, there was a correspondence between the stereochemistry of amino alcohols and the helical handedness of

polymers. The amino alcohols with stereochemistry of $S$ ($2^S$, $3^S$, and $4^S$) afforded the polymers with right-handed helicity (Fig. 4a, c, e), while those with stereochemistry of $R$ ($2^R$, $3^R$, and $4^R$) afforded the polymers with left-handed helicity (Fig. 4b, d, f). This correspondence implies that the supramolecular polymerization mechanism, as proposed in Fig. 2h, is generalized for these salts. Meanwhile, from these AFM images thus obtained, it was difficult to tell whether the supramolecular polymers adopt a double-strand helical structure or not, because AFM could visualize only the upper surface morphologies of the specimens[41].

**Stereoselective supramolecular copolymerization of two kinds of monomers**. Having obtained a family of supramolecular polymers with known helical handedness at hand, we then investigated the copolymerizability between different monomers[42]. Here, we fixed $1 \cdot 2^S$ as one of the mixing components, while its counterpart was chosen from $1 \cdot 3^S$, $1 \cdot 3^R$, $1 \cdot 4^S$, and $1 \cdot 4^R$.

We attempted the copolymerization of a homo-chiral combination, $1 \cdot 2^S$ and $1 \cdot 3^S$, by heating their dodecane solution ([$1 \cdot 2^S$] = [$1 \cdot 3^S$] = 2.0 mM) up to 348 K and then cooling it down to 283 K, where the CD spectrum was taken at 5-K step (Supplementary Fig. 13a, upper). The final observed CD spectrum was distinguishable from the calculated CD spectrum by averaging the spectra of separately prepared homopolymers of $1 \cdot 2^S$ and $1 \cdot 3^S$ (Fig. 5a), indicating that $1 \cdot 2^S$ and $1 \cdot 3^S$ interacted with each other to form a copolymer[43–46]. The less sigmoidal shape of the CD-monitored cooling curve suggested that the copolymerization proceeded via a cooperative mechanism with the elongation temperature ($T_e$) of 335 K (Fig. 5c, purple and Supplementary Fig. 13a, middle), which coincides with $T_e$ of the homopolymerization of $1 \cdot 2^S$ at 335 K (Fig. 5c, blue). We also confirmed that the structure of this copolymer was hardly influenced by the preparation process, because the association and dissociation of the monomer from the polymer was under fast equilibrium (Fig. 5e, i). When separately prepared homopolymers of $1 \cdot 2^S$ and $1 \cdot 3^S$ in dodecane were mixed at 283 K, the monomer exchange was completed within 1 min to yield the copolymer of $1 \cdot 2^S$ and $1 \cdot 3^S$ (Fig. 5g), whose final CD spectrum is perfectly identical to that of the copolymer prepared by cooling from a high temperature (Fig. 5h).

In contrast to the abovementioned homo-chiral combination, the hetero-chiral combination of $1 \cdot 2^S$ and $1 \cdot 3^R$ could not copolymerize. When their copolymerization was attempted using the procedure described above, the final CD spectrum was very close to the spectrum calculated by averaging those of the homopolymers of $1 \cdot 2^S$ and $1 \cdot 3^R$ (Fig. 5b), indicating that $1 \cdot 2^S$ and $1 \cdot 3^R$ were sorted to form segregated homopolymers[32,47–51]. Meanwhile, the cooling curve of the mixture, monitored with CD intensity at 354 nm, apparently showed a single elongation temperature ($T_e$) at ~320 K (Fig. 5d, purple and Supplementary Fig. 13b, middle). The shape of the curve was essentially unchanged when the monitoring wavelength was switched from 354 nm (Supplementary Fig. 13b, middle) to 325 nm (Supplementary Fig. 13b, lower), which verified that the homopolymers of $1 \cdot 2^S$ and $1 \cdot 3^R$ started growing almost simultaneously at ~320 K; note that CD signals at 354 nm were from both of the homopolymers of $1 \cdot 2^S$ and $1 \cdot 3^R$ (Fig. 3a, b), while those at 310 nm were solely from the homopolymer of $1 \cdot 3^R$ (Supplementary Fig. 14a, b). The elongation temperatures ($T_e$) of $1 \cdot 2^S$ and $1 \cdot 3^R$ in the mixed state (~320 K for both; Fig. 5d, purple) are notably lower than those of $1 \cdot 2^S$ and $1 \cdot 3^R$ in the unmixed state (335 and 325 K; Fig. 5d, blue and light red), suggesting that the salt-pairs of $1 \cdot 2^S$ and $1 \cdot 3^R$ mutually interfered in the nucleation stage to suppress their polymer elongation (Fig. 5f, nucleus, middle).

The stereoselective copolymerizability of $1 \cdot 2^S$ with $1 \cdot 3^S$ and $1 \cdot 3^R$ (Fig. 5e, f) was further confirmed by various spectroscopic approaches. Irrespective of the mixing ratio of monomers, the homo-chiral combination ($1 \cdot 2^S$ and $1 \cdot 3^S$) afforded CD spectra largely different from the linear sums of the spectra of unmixed monomers (Supplementary Fig. 15), while the spectra of the hetero-chiral combination ($1 \cdot 2^S$ and $1 \cdot 3^R$) were quite similar to the linear sums of the unmixed spectra (Supplementary Fig. 16). Furthermore, in the FT-IR measurement upon cooling from 348 to 298 K, the homo-chiral combination ($1 \cdot 2^S$ and $1 \cdot 3^S$) exhibited a characteristic absorption at 1568 cm$^{-1}$ (Supplementary Methods and Supplementary Fig. 17a) attributable to the hydrogen-bonded R-CO$_2^-$ group generated by the interaction between $1 \cdot 2^S$ and $1 \cdot 3^S$, while the hetero-chiral combination ($1 \cdot 2^S$ and $1 \cdot 3^R$) did not show the corresponding absorption (Supplementary Fig. 17b).

Likewise, the copolymerizability of other homo-chiral ($1 \cdot 2^S$ and $1 \cdot 4^S$) and hetero-chiral ($1 \cdot 2^S$ and $1 \cdot 4^R$) combinations was also investigated, where homo-chiral $1 \cdot 2^S$ and $1 \cdot 4^S$ interacted to form a copolymer[42–46], while hetero-chiral $1 \cdot 2^S$ and $1 \cdot 4^R$ were sorted to form segregated homopolymers (Supplementary Figs. 18 and 19)[32,47–51]. Considering the helical handedness of these polymers, we can establish a simple rule: two polymers with the same helical handedness are miscible ($1 \cdot 2^S$ [right-handed] + $1 \cdot 3^S$ [right-handed]; $1 \cdot 2^S$ [right-handed] + $1 \cdot 4^S$ [right-handed]), while those with opposite handedness are immiscible ($1 \cdot 2^S$ [right-handed] + $1 \cdot 3^R$ [left-handed]; $1 \cdot 2^S$ [right-handed] + $1 \cdot 4^R$ [left-handed]). Of further interest, the miscible combinations afforded helical fibrous copolymers (e.g., $1 \cdot 2^S$ + $1 \cdot 3^S$ [right-handed]; $1 \cdot 2^S$ + $1 \cdot 4^S$ [right-handed]) with the handedness similar to the homopolymers of their constituents, as confirmed by their AFM images (Supplementary Fig. 20). We believe the reason for this correspondence is because these supramolecular polymers adopted a similar hydrogen-bonded helical columnar network (Fig. 2h), where two kinds of salts can coexist in the same network only when their preferred helical handedness is the same.

Consistent with this rule, the similar CD measurements as described above revealed that homo-chiral $1 \cdot 3^S$ and $1 \cdot 4^S$ interacted to form a copolymer[42–46], while hetero-chiral $1 \cdot 3^S$ and $1 \cdot 4^R$ were sorted to form segregated homopolymers (Supplementary Fig. 21)[32,47–51]. Furthermore, we confirmed that the polymers of $1 \cdot 2^S$ and $1 \cdot 2^R$ were also immiscible[28–32], which was proved by the linear change of CD intensity with changing the ratio of $1 \cdot 2^S$ and of $1 \cdot 2^R$ in their mixture ($1 \cdot 2^S$:$1 \cdot 2^R$ = 100:0–0:100; Supplementary Fig. 22a), together with the fact that the elongation temperatures of enantiopure samples ($1 \cdot 2^S$:$1 \cdot 2^R$ = 100:0 and $1 \cdot 2^S$:$1 \cdot 2^R$ = 0:100; $T_e$ = 335 K for both) were notably higher than that of a racemic sample ($1 \cdot 2^S$:$1 \cdot 2^R$ = 50:50; $T_e$ = 315 K) (Supplementary Fig. 22b, c)[32].

A remaining issue is which type of copolymerization (block, alternating, random, etc.) is dominant in the above homo-chiral combinations. We attempted to address this question by AFM measurements, choosing the copolymer of $1 \cdot 2^S$ and $1 \cdot 4^S$ as a representative combination because of the AFM visibility of their homo- and copolymers. Thus, we measured AFM images of the mixtures of $1 \cdot 2^S$ and $1 \cdot 4^S$ with various molar ratios (100:0, 50:50, 25:75, and 0:100) and then statistically analyzed half helical pitches of the supramolecular polymers in these mixtures (Supplementary Fig. 23a–d). If we suppose that the block copolymerization is dominant (Supplementary Fig. 23e–h), the pitches of $1 \cdot 2^S$:$1 \cdot 4^S$ = 25:75 should show bimodal distributions with two peaks at the values of the unmixed $1 \cdot 2^S$ (5.3 nm; Supplementary Fig. 23a) and unmixed $1 \cdot 4^S$ (4.0 nm; Supplementary Fig. 23d). Meanwhile, if we suppose the dominance of the alternating copolymerization (Supplementary Fig. 23i–l), the pitches of $1 \cdot 2^S$:$1 \cdot 4^S$ = 25:75 should show a bimodal distribution with two peaks at the values of $1 \cdot 2^S$:$1 \cdot 4^S$ = 50:50 (4.8 nm;

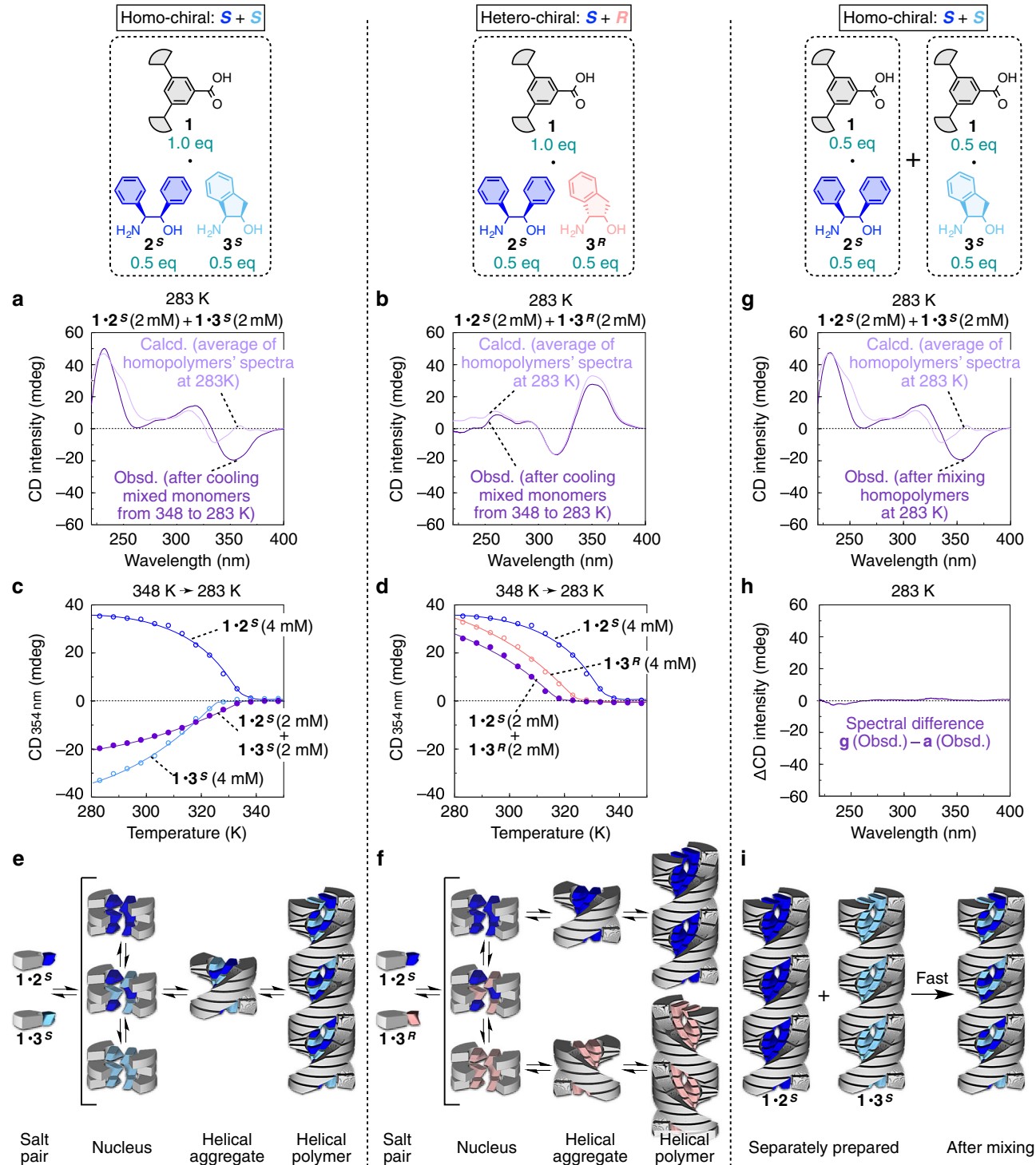

**Fig. 5 Stereoselective supramolecular copolymerization of two kinds of monomers. a** Obsd.: CD spectrum of a mixture of **1·2ˢ** and **1·3ˢ** in dodecane ([**1·2ˢ**] = [**1·3ˢ**] = 2.0 mM) after cooling from 348 to 283 K at −1.0 K/min. Calcd.: average of the CD spectra of **1·2ˢ** in dodecane (4.0 mM) and **1·3ˢ** in dodecane (4.0 mM) separately measured at 283 K. **b** Obsd.: CD spectrum of a mixture of **1·2ˢ** and **1·3ᴿ** in dodecane ([**1·2ˢ**] = [**1·3ᴿ**] = 2.0 mM) after cooling from 348 to 283 K at −1.0 K/min. Calcd.: average of the CD spectra of **1·2ˢ** in dodecane (4.0 mM) and **1·3ᴿ** in dodecane (4.0 mM) separately measured at 283 K. **c** Cooling curves (from 348 to 283 K at −1.0 K/min, monitored with CD intensity at 354 nm) of a mixture of **1·2ˢ** and **1·3ˢ** in dodecane ([**1·2ˢ**] = [**1·3ˢ**] = 2.0 mM), **1·2ˢ** in dodecane (4.0 mM), and **1·3ˢ** in dodecane (4.0 mM). **d** Cooling curves (from 348 to 283 K at −1.0 K/min, monitored with CD intensity at 354 nm) of a mixture of **1·2ˢ** and **1·3ᴿ** in dodecane ([**1·2ˢ**] = [**1·3ᴿ**] = 2.0 mM), **1·2ˢ** in dodecane (4.0 mM), and **1·3ᴿ** in dodecane (4.0 mM). **e**, **f** Schematics for the supramolecular polymerization of the homo-chiral (**1·2ˢ** and **1·3ˢ**; **e**) and hetero-chiral (**1·2ˢ** and **1·3ᴿ**; **f**) mixtures. **g** Obsd.: CD spectrum of an equi-volume mixture of **1·2ˢ** in dodecane (4.0 mM, before mixing) and **1·3ˢ** in dodecane (4.0 mM, before mixing) measured 30 s after mixing at 283 K. Calcd.: average of the CD spectra of **1·2ˢ** in dodecane (4.0 mM) and **1·3ˢ** in dodecane (4.0 mM) separately measured at 283 K. **h** Differential spectrum between the mixtures of **1·2ˢ** and **1·3ˢ** in **a** and **g**. **i** Schematic for merging of the supramolecular polymers of **1·2ˢ** and **1·3ˢ**.

Supplementary Fig. 23b) and unmixed $1 \cdot 4^S$ (4.0 nm; Supplementary Fig. 23d). However, the experimentally observed pitches of $1 \cdot 2^S{:}1 \cdot 4^S = 25{:}75$ showed unimodal narrow distributions at 4.3 nm (Supplementary Fig. 23c), excluding the possibility of the block and alternating copolymerization. Considering the fact that the pitches of the series of samples monotonically varied with changing the ratio of $1 \cdot 2^S$ and $1 \cdot 4^S$ (Supplementary Fig. 23a–d), the present copolymerization was likely to proceed in a random manner[42].

Overall, the result described in this section can be interpreted as follows: the polymer of $1 \cdot 2^S$ can recognize the chirality of the antipodes $3^S/3^R$ and $4^S/4^R$ based on the stereoselective copolymerization mechanism, where homo-chiral guests ($3^S$ and $4^S$) were accommodated while hetero-chiral guests ($3^R$ and $4^R$) were rejected. Such recognition ability of the polymer of $1 \cdot 2^S$ was extended to the enantio-separation of $3^S/3^R$ and $4^S/4^R$, as discussed in the next section.

**Enantio-separation of amino alcohols by the supramolecular polymer.** In all of the above experiments, supramolecular polymers were isotropically dissolved in dodecane, which was evidenced by negligible linear dichroism of these solutions. Indeed, the polymer of $1 \cdot 2^S$ ($1 \cdot 2^R$) in dodecane (4.0 mM) remained homogeneous and clear for several months (Supplementary Fig. 24a). Meanwhile, when the polymer of $1 \cdot 3^S$ ($1 \cdot 3^R$) in dodecane (4.0 mM) was kept at 298 K for a long time (e.g., 12 h), it started bundling to form super-helical assembles that were insoluble in dodecane (Supplementary Fig. 24b). In other words, the homogeneously dissolved state of the polymer of $1 \cdot 3^S$ ($1 \cdot 3^R$) was a sort of metastable state that could stand for several hours, but eventually turned into a more thermodynamically stable insoluble state. Similar solution instability was found for the polymer of $1 \cdot 4^S$ ($1 \cdot 4^R$) (Supplementary Fig. 24c).

Considering such a large difference in solution stability between the supramolecular polymers, together with their stereoselective copolymerizability, we anticipated that the polymer of $1 \cdot 2^S$ would serve as a "stereoselective solubilizer" for the supramolecular polymers of $1 \cdot 3^S$ and $1 \cdot 3^R$. When the polymers of $1 \cdot 2^S$ and $1 \cdot 3^R$, separately prepared in dodecane (4.0 mM), were mixed at 298 K in the same amount, the polymer of $1 \cdot 3^R$ did not react with $1 \cdot 2^S$, because of their hetero-chiral relationship. Consequently, the polymer of $1 \cdot 3^R$ turned into insoluble aggregates (Fig. 6b) because of its intrinsic solution instability (Supplementary Fig. 27b). In sharp contrast, when the similar mixing experiment was conducted between the polymers of $1 \cdot 2^S$ and $1 \cdot 3^S$, they merged immediately to form the copolymer of $1 \cdot 2^S$ and $1 \cdot 3^S$, because of their homo-chiral relationship (Fig. 5g–i). The resultant copolymer exhibited much higher solution stability than the polymer of $1 \cdot 3^S$, and sustained the homogeneously dissolved state (Fig. 6a).

We then conjectured such stereoselective solubilization ability of $1 \cdot 2^S$ would lead to the enantio separation of $3^S/3^R$ (enantio-separation of the amino alcohols protocol 1 in Methods and Fig. 6c). When separately prepared polymers in dodecane of $1 \cdot 3^S$ (4.0 mM, 0.84 μmol in 210 μL), $1 \cdot 3^R$ (4.0 mM, 0.84 μmol in 210 μL), and $1 \cdot 2^S$ (4.0 mM, 1.68 μmol in 420 μL) were sequentially mixed and kept at 298 K, the stereoselective copolymerization of $1 \cdot 2^S$ and $1 \cdot 3^S$ occurred to form a soluble copolymer of $1 \cdot 2^S$ and $1 \cdot 3^S$, while the polymer of $1 \cdot 3^R$ did not react with $1 \cdot 2^S$ and turned to insoluble aggregates. Owing to such solubility difference between the copolymer of $1 \cdot 2^S$ and $1 \cdot 3^S$ and the homopolymer of $1 \cdot 3^R$, enantio biasing of $3^S/3^R$ took place between the supernatant ($3^S$-enriched) and the precipitate ($3^R$-enriched) (Fig. 6c). Indeed, the CD spectrum of the supernatant was quite similar to that of a mixture of $1 \cdot 2^S$ and $1 \cdot 3^S$ (3:1, mol/mol) in dodecane (Supple-

mentary Fig. 25). From the supernatant, a $3^S$-enriched mixture (0.42 μmol, $3^S{:}3^R = 90{:}10$) was obtained in 49% yield with respect to the original amount of $3^S$ (chiral high-performance liquid chromatography (HPLC) in Methods and Fig. 6d). This enantio-separation could be performed in good reproducibility, in terms of both yield and selectivity (Supplementary Fig. 26a). Furthermore, $1$ could be quantitatively recovered from the supernatant and precipitate (reuse of the carboxylic acid in Methods) and used for the second cycle of the enantio-separation, where yield and selectivity were similar to the first cycle (Supplementary Fig. 26c).

Generality of the present enantio-separation, in terms of both selectors and guests, could be successfully proved by the following demonstrations. By using the same selector (polymer of $1 \cdot 2^S$) and the same procedure as described above (protocol 1; Supplementary Fig. 27c), the enantio-separation of another set of guests $4^S$/$4^R$ was also realized to afford a $4^S$-enriched mixture ($4^S{:}4^R = 95{:}5$) in 47% yield with respect to the original amount of $4^S$ (Fig. 6e); the enantio-enrichment degree of $4^S$ was higher than that of $3^S$, because the polymer of $1 \cdot 4^R$ showed lower solubility than that of $1 \cdot 3^R$ (Supplementary Fig. 24). As the selector, not only the polymer of $1 \cdot 2^S$ but also another polymer composed of $1 \cdot 5^S$ (Supplementary Fig. 28) and even the copolymer of $1 \cdot 2^S$ with $1 \cdot 5^S$ (Supplementary Figs. 29 and 30) successfully served for the separation of $3^S/3^R$ and $4^S/4^R$. One unique feature of the present enantio-separation is that the order of affinity between the selector and the guest enantiomers can be rationally predicted from the homo-chiral copolymerization rule, i.e., the selector accommodates the homo-chiral guest while rejects the hetero-chiral one.

Although the protocol in Fig. 6c (protocol 1) needs the pretreatment of the target guests ($3^S$ and $3^R$) to convert into the salts with $1$ (e.g., $1 \cdot 3^S$ and $1 \cdot 3^R$, respectively), the direct enantio-separation from racemic $3^S/3^R$ was realized by changing the mixing order of reagents (enantio-separation of the amino alcohols protocol 2 in Methods and Supplementary Fig. 31a). When a dodecane solution (1200 μL) containing $1 \cdot 2^S$ (3.0 mM, 3.60 μmol) and free acid $1$ (1.0 mM, 1.20 μmol) was mixed with a racemic solid of $3^S/3^R$ (0.87 μmol/0.87 μmol) at 298 K, $3^S$ and $3^R$ in the solid gradually dissolved into dodecane to form the salt-pairs $1 \cdot 3^S$ and $1 \cdot 3^R$, so that the enantio biasing of $3^S/3^R$ proceeded in the same mechanism as Fig. 6c. From the supernatant, a $3^S$-enriched mixture (0.62 μmol, $3^S{:}3^R = 88{:}12$) was obtained in 71% yield with respect to the original amount of $3^S$ (Supplementary Fig. 31b). Compared with the protocol in Fig. 6c (protocol 1) the new protocol in Supplementary Fig. 31a (protocol 2) realized a better yield with retaining almost similar selectivity, probably because the new protocol enabled the gradual generation of $1 \cdot 3^S$ and $1 \cdot 3^R$ in dodecane, so that monomeric $1 \cdot 3^S$ is mainly consumed for the copolymerization with $1 \cdot 2^S$ rather than its homo-polymerization. In protocol 2, the addition of free acid $1$ was critical for realizing high efficiency in enantio-separation; in the absence of free acid $1$ (enantio-separation of the amino alcohols protocol 3 in Methods and Supplementary Fig. 32a), only small enantio-enrichment took place (Supplementary Fig. 32c), probably because dissolution of guest molecules in dodecane was supressed.

## Discussion

The helical supramolecular polymer developed in this work realized the enantio-separation of chiral small molecules, in which a simple one-pot process could resolve racemic mixtures into enantio-enriched materials. The key to this achievement is our strategy of using the non-covalent interaction sites in the supramolecular polymer not only for the connection of the monomers but also for the recognition of chiral guests. It is worth noting is

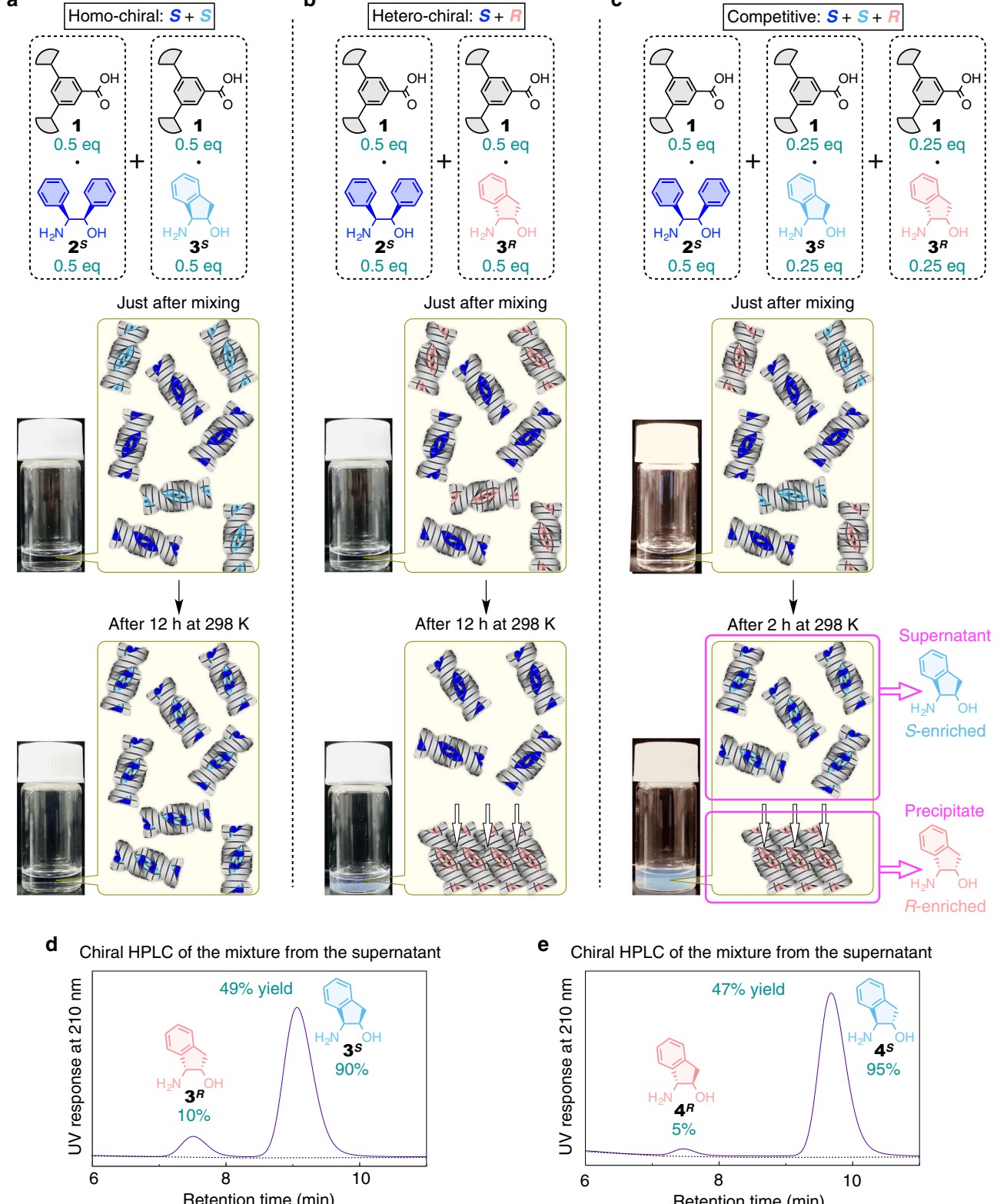

**Fig. 6 Enantio-separation of amino alcohols by the helical supramolecular polymer. a–c** Photographs and schematics of the supramolecular polymers prepared at 298 K by mixing separately prepared dodecane solutions (4.0 mM) of **1·2^S** (420 μL) and **1·3^S** (420 μL) (**a**), **1·2^S** (420 μL) and **1·3^R** (420 μL) (**b**), and **1·2^S** (420 μL), **1·3^S** (210 μL), and **1·3^R** (210 μL) (**c**). Just after mixing (upper) and after incubation at 298 K (lower). **d** Chiral HPLC trace of the mixture of **3^S** and **3^R** obtained from the supernatant of **c**. **e** Chiral HPLC trace of the mixture of **4^S** and **4^R** obtained from the supernatant of Supplementary Fig. 27c. For details, see Methods.

that similar enantio-separation ability should reside in all reported supramolecular polymers that are formed via homochiral polymerization[28–32], but has never been investigated. Speaking not only chirality recognition but general molecular recognition, all supramolecular polymers with self-sorting nature also potentially have the similar separation ability[47–51]. Considering the great success of chiral selectors and catalysts based on helical covalent polymers[5–8], helical supramolecular polymers with such chirality recognition ability would also find unique applications, taking advantage of their dynamic nature and stimuli-responsive characteristics[7].

In addition to the strategy described above, another key to the present achievement is the particular design of our supramolecular polymer, whose monomer consists of an achiral main body and a non-covalently attached chiral auxiliary. As the chiral auxiliary is from a "chiral pool" of amino alcohols[34], a library of chiral supramolecular polymers can be readily formed. Before this work, there were only a few helical supramolecular polymer systems that can non-covalently accommodate various kinds of chiral auxiliaries from chiral pools[12,14,19–21]. Our systematic studies revealed the clear relationship between the stereochemistry of the chiral auxiliaries, helical handedness of the polymers, and copolymerizability of the polymers. Such a library of helical supramolecular polymers with well-understood structures and properties will further expand the possibility of supramolecular polymers.

## Methods

**Enantio-separation of the amino alcohols protocol 1**. Separately prepared dodecane solutions (4.0 mM) of $1 \cdot 3^S$ (210 μL), $1 \cdot 3^R$ (210 μL), and $1 \cdot 2^S$ (420 μL) were sequentially added to a glass vial at 298 K and left to stand at 298 K for 2 h, where a precipitate formed. The resultant suspension was subjected to centrifugation, and the supernatant was extracted with 1000 μL of HClO$_4$ (pH 2.0)/MeOH = 80/20, v/v. The aqueous layer was directly injected to chiral HPLC (see below) to determine the enantiomeric composition and recovery yield of the mixture of $3^S$ and $3^R$ from the HPLC peak areas. Chiral separation of $4^S$ and $4^R$ was conducted in the same procedure.

**Enantio-separation of the amino alcohols protocol 2**. A CHCl$_3$ solution (218 μL) of $3^S$ (4.0 mM, 0.87 μmol) and a CHCl$_3$ solution (218 μL) of $3^R$ (4.0 mM, 0.87 μmol) was added to a glass vial, and the solvent was slowly evaporated to dryness under ambient conditions. To the glass vial containing solid $3^S/3^R$, a dodecane solution (1200 μL) containing $1 \cdot 2^S$ (3.0 mM, 3.60 μmol) and free acid $1$ (1.0 mM, 1.20 μmol) was added at 298 K. The mixture was left to stand at 298 K for 1.3 h, where the mixture once became homogeneous, and then precipitate formed. The resultant suspension was subjected to centrifugation, and the supernatant was extracted with 4000 μL of HClO$_4$ (pH 2.0)/MeOH = 80/20, v/v. The aqueous layer was directly injected to chiral HPLC (see below) to determine the enantiomeric composition and recovery yield of the mixture of $3^S$ and $3^R$ from the HPLC peak areas.

**Enantio-separation of the amino alcohols protocol 3**. A CHCl$_3$ solution (200 μL) of $3^S$ (4.0 mM, 0.80 μmol) and a CHCl$_3$ solution (200 μL) of $3^R$ (4.0 mM, 0.80 μmol) was added to a glass vial, and the solvent was slowly evaporated to dryness under ambient conditions. To the glass vial containing solid $3^S/3^R$, a dodecane solution (400 μL) containing $1 \cdot 2^S$ (4.0 mM, 1.60 μmol) was added at 298 K. The mixture was left to stand at 298 K for 1.3 h, where the mixture once became homogeneous, and then precipitate formed. The resultant suspension was subjected to centrifugation, and the supernatant was extracted with 4000 μL of HClO$_4$ (pH 2.0)/MeOH = 80/20, v/v. The aqueous layer was directly injected to chiral HPLC (see below) to determine the enantiomeric composition and recovery yield of the mixture of $3^S$ and $3^R$ from the HPLC peak areas. Enantio-separation of $4^S$ and $4^R$ was conducted in the same procedure.

**Chiral HPLC**. Chiral HPLC analysis of the mixture of $3^S$ and $3^R$ was performed on a JASCO model PU-4080i pump equipped with a model UV-4075 detector. Column: Daicel CHIRALCEL CROWNPAK CR(+) (4.0 × 150 mm). Eluent: pH 2.0 aqueous HClO$_4$/MeCN = 95/5, v/v. Temperature: 273 K. Flow rate: 0.60 mL/min. Injection volume: 5.0 μL. Detection: UV absorption at 210 nm. Elution time: 7.5 min ($3^R$) and 9.1 min ($3^S$). Chiral HPLC analysis of the mixture of $4^S$ and $4^R$ was performed under the same conditions as those of the case of $3^S$ and $3^R$. Elution time: 7.5 min ($4^R$) and 9.7 min ($4^S$).

**Reuse of the carboxylic acid**. The first run of the enantio-separation was conducted following protocol 1 as described above. Thus, separately prepared dodecane solutions (4.0 mM) of $1 \cdot 3^S$ (210 μL), $1 \cdot 3^R$ (210 μL), and $1 \cdot 2^S$ (420 μL) were mixed and left to stand at 298 K for 2 h, where a precipitate formed. The supernatant and precipitate were separated by centrifugation. The supernatant was washed with 2000 μL of HClO$_4$ (pH 2.0)/MeOH = 80/20, v/v, while the precipitate was suspended in 600 μL of dodecane and washed with 2000 μL of HClO$_4$ (pH 2.0)/MeOH = 80/20, v/v. The dodecane layers were combined and dried under reduced pressure (~1 mmHg) at 298 K for 24 h to afford a dodecane suspension of $1$ (~3.2 μmol). The suspension was mixed with a CHCl$_3$ solution of $2^S$ (4.0 mM, 800 μL), concentrated under reduced pressure (~1 mmHg) at 298 K for 48 h, and diluted with dodecane to adjust the total volume to 800 μL to afford a dodecane solution of $1 \cdot 2^S$ (4.0 mM), which was used for the second run of the enantio-separation. For further information on methods, see Supplementary Methods.

## Data availability

The authors declare that the data supporting the findings of this study are available within the paper and its supplementary information file. All other information is available from the corresponding authors upon reasonable request.

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

## Acknowledgements
This work was financially supported by JST CREST Grant Number JPMJCR17N1, Japan. We thank D. Miyajima (RIKEN) and K. Sugiyasu (National Institute for Materials Science) for discussion and technical advices. We thank M. Kawamoto (RIKEN) for providing an FT-IR instrument.

## Author contributions
V.K.P., A.A.S., and Y.I. conceived the project; K.S. designed and performed all experiments. V.K.P. and A.A.S. co-designed experiments; K.S., A.A.S., and K.Y. synthesized compounds; N.N.H. measured AFM data, and Y.I. analyzed them; K.S., V.K.P., A.A.S., and Y.I. co-wrote the paper.

## Competing interests
The authors declare no competing interests.
