## [Peer Review File · Nature Communications]

Editorial Note: The figure on page 4 of this Peer Review File is reproduced with permission from Würthner, F. et al. (J. Am. Chem. Soc. 137, 3300 (2015)). Copyright (2015) American Chemical Society.

Reviewers' comments:

Reviewer #1 (Remarks to the Author):

This paper concerns the chiral additives (different kinds of amino alcohol) triggered self-assembly of achiral molecules (dendritic carboxylic acid). As expected, the helical handedness of the assembled polymers is determined by the stereochemistry of the chiral molecules. This phenomenon has been reported for many years, and you can find numerous examples with similar results. The authors just described another example in the first half of the main text. In the rest part, when polymers with same chirality were mixed, copolymerization was proved by using temperature-dependent characterizations. Based on this observation, the authors showed the chiral recognition rather than "enantio-separation" among different chiral polymers. The figures are complicated, and too many molecular structures are listed although some of them are repeated. I do not think that, with inclusion of the points I raise, the manuscript is suitable for the aim of Nat. Commun. Any other more specialized journal can go ahead.

1. It should be noted that the word "enantio-separation" used in manuscript is obviously inaccurate, because the mixture of 3S/3R was not separated. Actually, it's just a chiral recognition process, only the polymers of 1-3S or 1-3R was distinguished, while the molecules of 3S and 3R still existed in the polymers. "Enantio-separation" should be removed from the text if no additional improvements were provided.
2. As far as helical assembly is considered, the authors have given the cartoon illustration in Figure 1 and other Figures, which is something like double-strand helix. However, from the morphology observation, there appeared single-strand helix both in AFM and in TEM images. What about the relationship between the observed results and the illustrated patterns?
3. The investigation of 1-3S with 1-4S and 1-4R should be illustrated. Does copolymerization occur in these cases? Corresponding explanations are needed.
4. Is it possible to recognize and separate 3S/3R or 4S/4R directly by 1-2S without further addition of 1?
5. Why the selectivity is different for the separation of 3S/3R and 4S/4R, although the structure of 3 is very similar to that of 4?
6. The selectivity and the yield for the chiral separation are not good enough.
7. The number of Figures, especially those in supplementary part, should follow the order of their appearance in the main text. It's really confused for the readers with disorder figures.
8. The words and sentences used in this paper seem arbitrary. The following statements are dangerous from a scientific point of view: "So far, only a few examples of chiral reactions have been reported"; "while there has been no report on enantio-separation"; "This idea has never been explored but seems reasonable"... I strongly suggest the authors reread the seminal papers in this area.

Reviewer #2 (Remarks to the Author):

The authors designed and synthesized a helical polymer with chiral sites and this chirality determines the helicity of the polymer. They claimed that this chiral polymer can easily and effectively separate enantiomers of other chiral molecules. Their molecular and experiment design is elaborate and careful, and the reviewer believes that the novelty of the paper is high enough to be published in Nature Communication. Below are some points that reviewer noticed and strongly recommends to be considered and resolved prior to publication.

1. On page 4, "salt - pair interaction" is a confusing term to use, it is either electrostatic interaction, hydrogen bonding or any kind of physical interactions; besides, it is better to choose one interaction in the rest of the paper, readers might be confused if it is electrostatic interaction triggering polymerization or H - bonding.
2. Authors claimed that the helical polymer 1-2S can separate the enantiomers of chiral molecules,

so I suggest them to provide CD spectra of the enantiomer which is excluded by the polymer in addition to provided chiral HPLC data(e.g. 3S), to prove the purity of chiral compound left in solution (or probably precipitated).

3. It is highly recommended to indicate Elongation Temperatures on Figures 2 and 3.

4. Authors claimed that the helical polymers with the same chirality (e.g. 1-2S and 1-3S) copolymerize and suggested a structure for the copolymer, but their only evidence was that observed CD spectra of mixture did not match with the calculated CD spectra of the average of two polymer. This reviewer believes this is not a strong evidence, and instead of the structure authors suggested, two polymers might form any other kind of structure due to their interaction.

Reviewer #3 (Remarks to the Author):

This manuscript reported the application of a helical supramolecular polymer to the enantio-separation of chiral guest molecules. By using the strategy that the non-covalent interaction sites in the supramolecular polymer can not only connected the monomers but also recognized the chiral guest molecules, the authors designed a new class of helical supramolecular polymers, whose monomer is composed of an achiral main body and a chiral auxiliary. The fundamental properties of these helical supramolecular polymers and their copolymerizability between different monomers as well as the chirality recognition ability were carefully studied. Based on the stereoselective copolymerization mechanism, these helical supramolecular polymers can recognize the chirality of the antipode molecules, where homo-chiral guests were accommodated while hetero-chiral guests were rejected. In a simple one-pot process, these helical supramolecular polymers could resolve racemic mixtures into highly enantio-enriched materials in good yields. This study, to a certain extent, revealed the structure-property relationship between the stereochemistry of the chiral auxiliaries, helical handedness of the polymers, and copolymerizability of the polymers. The experimental part of this work is well prepared with consistent data, however, publication of this work on Nature Communications is recommended after some improvements below:

(1) The elongation temperatures (T_e) were measured for these supramolecular polymers and the values were various from each other, the relationships between the structure of the supramolecular polymers and T_e values should be studied in this work.

(2) Whether random copolymerization or other copolymerization method is adopted for monomers with the homo-chiral property? And do the obtained supramolecular copolymers lead to the enantio-separation of enantiomers?

(3) For the enantio-separation of 3S/3R, the author used two different mixing order of reagents, and got 3S-enriched mixture with different S:R ratios and yields, The factors that are relevant to the separation efficiency and yield should be carefully discussed.

(4) Does this family of helical supramolecular polymers can lead to the completely enantio-separation of relevant chiral guest molecules? What about the advantages over other methods of enantio-separation?

For Reviewer #1

[0] This paper concerns the chiral additives (different kinds of amino alcohol) triggered self-assembly of achiral molecules (dendritic carboxylic acid). As expected, the helical handedness of the as-formed polymers is determined by the stereochemistry of the chiral molecules. This phenomenon has been reported for many years, and you can find numerous examples with similar results. The authors just described another example in the first half of the main text. In the rest part, when polymers with same chirality were mixed, copolymerization was proved by using temperature-dependend characterizations. Based on this observation, the authors showed the chiral recognition rather than “enantio-separation” among different chiral polymers. The figures are complicated, and too many molecular structures are listed although some of them are repeated. I do not think that, with inclusion of the points I raise, the manuscript is suitable for the aim of *Nat. Commun.* Any other more specialized journal can go ahead.

=> We appreciate these valuable comments, as well as constructive suggestions of additional experiments. We conducted all of them, which certainly reinforced our claim.

=> We are afraid that the reviewer overlooked our demonstration of the enantio-separation of a racemic guest (see our answer to comment [1]), which largely affected his/her evaluation. This misunderstanding might be caused by unclear descriptions in our previous manuscript, and therefore we rewrote the corresponding parts in the present revision.

=> This work firstly proved that the monomer-connection sites in helical supramolecular polymers are useful for the recognition of chiral small molecules. We hope the reviewer could kindly understand this point.

=> We showed molecular formulas for all of the illustration panels, because this work deals with various amino alcohols with resembling structures. Judging from the comments of other reviewers, they seem to be satisfied with our style of representation. We would like to leave the decision on this issue to the editor.

[1] It should be noted that the word “enantio-separation” used in manuscript is obviously inaccuracy, because the mixture of $3^S/3^R$ was not separated. Actually, it's just a chiral recognition process, only the polymers of $1\cdot 3^S$ or $1\cdot 3^R$ was distinguished, while the molecules of 3^S and 3^R still existed in the polymers. “Enantio-separation” should be removed from the text if no additional improvements were provided.

=> We are afraid that the reviewer overlooked our result on the direct enantio-separation from a racemic mixture $3^S/3^R$ (71% yield, $3^S:3^R = 88:12$; Supplementary Fig. 26). For experimental details, please see Supplementary Method 7, protocol 2.

=> To avoid such misunderstanding, we rewrote the corresponding part in the main text (page 15, line 3–4).

[2] As far as helical assembly is considered, the authors have given the cartoon illustration in Fig. 1 and other Figs, which is something like double-strand helix. However, from the morphology observation, there appeared single-strand helix both in AFM and in TEM images. What about the relationship between the observed results and the illustrated patterns?

=> First of all, we did not use TEM at all in this work. The following is discussion on AFM data.

=> Because AFM visualizes only the upper-surface morphology of the observed object, it is generally difficult to distinguish single-strand and double-strand helices from AFM images.

=> For your quick understanding, here we introduce an example of a helical supramolecular polymer reported by Würthner, F. *et al.* (*J. Am. Chem. Soc.* **137**, 3300 (2015)). The alkyl-chain parts in the monomer are supposed to adopt a double-strand helical structure (Fig. 1a in below), while it is difficult to tell from the AFM image whether the helix is single- or double-strand (Fig. 1c in below).

Figure 1.

a, Chemical structure of **PBI-1** used for the studies on the seeded polymerization, and schematic representation of the hydrogen-bond-directed self-assembly behavior of **PBI-1**.

b, Temperature-dependent absorption spectral changes of **PBI-1** in MCH/toluene (2:1, v/v) observed during cooling process from 363 K (pink line) to 303 K (blue line). Condition: $c_T = 1.0 \times 10^{-5} \text{ M}$.

c, Left top, AFM height image (1.5 \times 1.5 μm , the z scale is 10 nm) of **PBI-1_Agg**, spin-coated (3000 rpm) onto silicon wafers from the MCH/toluene solutions. Left bottom, cross-section analysis corresponding to the yellow dashed line. Right, height image (0.46 \times 0.46 μm , the z scale is 10 nm) of the region enclosed with the black line.

=> To explain this issue, we added a description to the main text (page 9, line 17–19) with citing the above paper (ref. 41). In addition, to avoid possible confusion, we changed the illustrations of helices adjacent to the AFM images in the main text (Fig. 4) and in Supplementary Information (Supplementary Fig. 11 / Supplementary Fig. 16).

- [3] The investigation of $1\cdot 3^S$ with $1\cdot 4^S$ and $1\cdot 4^R$ should be illustrated. Does copolymerization occur in these cases? Corresponding explanations are needed.
- => We investigated the copolymerizability of $1\cdot 3^S$ with $1\cdot 4^S$ and $1\cdot 3^S$ with $1\cdot 4^R$. As expected, the copolymerization occurred only in the homo-chiral combination.
- => To provide this information, we added a description to the main text (page 11, line 21–23) and the corresponding data to Supplementary Information (Supplementary Fig. 17).
- [4] Is it possible to recognize and separate $3^S/3^R$ or $4^S/4^R$ directly by $1\cdot 2^S$ without further addition of **1**?
- => We conducted the enantio-separation of $3^S/3^R$ or $4^S/4^R$ by $1\cdot 2^S$ without adding free acid **1**. In both cases, enantio-enrichment took place, but the efficiency was not sufficient (Supplementary Fig. 27).
- => To provide this information, we added a description to the main text (page 15, line 16–20) and the corresponding description and data to Supplementary Information (Supplementary Method 7, protocol 3 and Supplementary Fig. 27).
- [5] Why the selectivity is different for the separation of $3^S/3^R$ and $4^S/4^R$, although the structure of **3** is very similar to that of **4**?
- => This enantio-separation is based on the lower solubility of the homopolymer of $1\cdot 3^R$ (or $1\cdot 4^R$) than that of the copolymer of $1\cdot 2^S$ and $1\cdot 3^S$ (or $1\cdot 2^S$ and $1\cdot 4^S$). As shown in Supplementary Fig. 20, the homopolymer of $1\cdot 4^R$ shows lower solubility than that of $1\cdot 3^R$. Accordingly, in the enantio-separation process, the remaining amount of 4^R in the supernatant is lower than that of 3^R , which leads to higher selectivity in the separation of $4^S/4^R$ than that of $3^S/3^R$.
- => To provide this explanation, we added a brief description to the main text (page 14, line 18–20).
- [6] The selectivity and the yield for the chiral separation are not good enough.
- => The core of this work is to show the potential of helical supramolecular polymer as a selector for recognizing the chirality of guest molecules. The current selectivity and yield are not perfect but enough to prove our concept. We would like to improve the selectivity and yield in follow-up works.
- => Please also note that Reviewer #3 evaluated the current selectivity and yield as follows: “... these helical supramolecular polymers could resolve racemic mixtures into highly enantio-enriched materials in good yields”.
- => Because the evaluation criteria depends on the person, we refrained from using the phrases ‘high efficiency’ and ‘good yield’ in the revised main text (page 2, line 12 / page 4, line 17–18 / page 16, line 3–4).

[7] The number of Figures, especially those in supplementary part, should follow the order of their appearance in the main text. It's really confused for the readers with disorder figures.

=> In response to this kind suggestion, we renumbered the figures in Supplementary Information.

[8] The words and sentences used in this paper seem arbitrary. The following statements are dangerous from a scientific point of view: "So far, only a few examples of chiral reactions have been reported"; "while there has been no report on enantio-separation"; "This idea has never been explored but seems reasonable"... I strongly suggest the authors reread the seminal papers in this area.

=> In response to this kind suggestion, we rewrote the corresponding parts in the main text (page 3, line 9 / page 3, line 10–11 / page 3, line 22).

For Reviewer #2

[0] The authors designed and synthesized a helical polymer with chiral sites and this chirality determines the helicity of the polymer. They claimed that this chiral polymer can easily and effectively separate enantiomers of other chiral molecules. Their molecular and experiment design is elaborate and careful, and the reviewer believes that the novelty of the paper is high enough to be published in *Nature Communications*. Below are some points that reviewer noticed and strongly recommends to be considered and resolved prior to publication.

=> We appreciate these highly encouraging comments, as well as constructive suggestions of additional experiments. We conducted all of them, which certainly reinforced our claim.

[1] On page 4, “salt–pair interaction” is a confusing term to use, it is either electrostatic interaction, hydrogen bonding or any kind of physical interactions; besides, it is better to choose one interaction in the rest of the paper, readers might be confused if it is electrostatic interaction triggering polymerization or H–bonding.

=> We appreciate the reviewer for this important comment. We rephrased the term “salt–pair interaction” with appropriate expressions (page 4, line 4).

[2] Authors claimed that the helical polymer $1 \cdot 2^S$ can separate the enantiomers of chiral molecules, so I suggest them to provide CD spectrum of the enantiomer which is excluded by the polymer in addition to provided chiral HPLC data (e.g. 3^S), to prove the purity of chiral compound left in solution (or probably precipitated).

=> We guess the reviewer suggested measuring the CD spectrum of the recovered guest after isolation from **1**. However, it is technically difficult because the large CD signals at the visible region are observed only when the guest forms a helical supramolecular polymer with **1** bearing a conjugated aromatic system.

=> Instead, we measured the CD spectrum of the supernatants in the enantio-separation of $3^S/3^R$ by $1 \cdot 2^S$, which was close to the CD spectrum of an authentic sample of the copolymer of $1 \cdot 2^S$ and $1 \cdot 3^S$ at a molar ratio of ca. 3:1. This observation further supports the enrichment of 3^S in the supernatant through our proposed mechanism.

=> To provide this information, we added a description to the main text (page 14, line 9–11) and the corresponding data to Supplementary Information (Supplementary Fig. 21).

[3] It is highly recommended to indicate elongation temperatures on Figs 2 and 3.

=> In response to this kind suggestion, we indicated the elongation temperatures in the suggested figures in the main text (Figs 2 and 3).

- [4] Authors claimed that the helical polymers with the same chirality (e.g. $1\cdot 2^S$ and $1\cdot 3^S$) copolymerize and suggested a structure for the copolymer, but their only evidence was that observed CD spectra of mixture did not match with the calculated CD spectra of the average of two polymer. This reviewer believes this is not a strong evidence, and instead of the structure authors suggested, two polymers might form any other kind of structure due to their interaction.
- => We really appreciate this important comment. We measured AFM images of an equimolar mixture of $1\cdot 2^S$ with $1\cdot 3^S$ and that of $1\cdot 2^S$ with $1\cdot 4^S$. In both cases, supramolecular fibres with the same helical handedness as their constituent homopolymers (= right-handed) were observed.
- => To provide this information, we added a description to the main text (page 11, line 14–17) and the corresponding data to Supplementary Information (Supplementary Fig. 16).

For Reviewer #3

[0] This manuscript reported the application of a helical supramolecular polymer to the enantio-separation of chiral guest molecules. By using the strategy that the non-covalent interaction sites in the supramolecular polymer can not only connected the monomers but also recognized the chiral guest molecules, the authors designed a new class of helical supramolecular polymers, whose monomer is composed of an achiral main body and a chiral auxiliary. The fundamental properties of these helical supramolecular polymers and their copolymerizability between different monomers as well as the chirality recognition ability were carefully studied. Based on the stereoselective copolymerization mechanism, these helical supramolecular polymers can recognize the chirality of the antipode molecules, where homo-chiral guests were accommodated while hetero-chiral guests were rejected. In a simple one-pot process, these helical supramolecular polymers could resolve racemic mixtures into highly enantio-enriched materials in good yields. This study, to a certain extent, revealed the structure-property relationship between the stereochemistry of the chiral auxiliaries, helical handedness of the polymers, and copolymerizability of the polymers. The experimental part of this work is well prepared with consistent data, however, publication of this work on *Nature Communications* is recommended after some improvements below:

=> We appreciate these highly encouraging comments, as well as constructive suggestions of additional experiments. We conducted all of them, which certainly reinforced our claim.

[1] The elongation temperatures (T_e) were measured for these supramolecular polymers and the values were various from each other, the relationships between the structure of the supramolecular polymers and T_e values should be studied in this work.

=> We anticipated that the conformational stability of the amino alcohol units would be one of the most determinant structural parameters for the elongation temperatures (T_e) of the supramolecular polymers of $\mathbf{1}\cdot\mathbf{2}^S$ ($T_e = 335$ K), $\mathbf{1}\cdot\mathbf{3}^S$ ($T_e = 325$ K) and $\mathbf{1}\cdot\mathbf{4}^S$ ($T_e = 345$ K). Our survey of the Cambridge Structural Database (CSD) revealed that the order of conformational stability of the amino alcohols is $4^S > 2^S > 3^S$, which is consistent with the order of T_e .

=> To provide this information, we added a description to the main text (page 8, line 4–25) and the corresponding data to Supplementary Information (Supplementary Fig. 9).

[2-1] Whether random copolymerization or other copolymerization method is adopted for monomers with the homo-chiral property?

=> To address this important comment in an unambiguous manner, we conducted direct visualization of the homo-chiral copolymer by AFM. We chose the copolymer of $\mathbf{1}\cdot\mathbf{2}^S$ and $\mathbf{1}\cdot\mathbf{4}^S$ as a representative, because their homopolymers and copolymers could be clearly visualized by AFM as helical fibres.

=> We prepared the specimens by mixing $\mathbf{1}\cdot\mathbf{2}^S$ and $\mathbf{1}\cdot\mathbf{4}^S$ at various molar ratios of 100:0, 50:50, 25:75, and 0:100. In the AFM image of each specimen, helices with a unimodal narrow pitch distribution were observed, which exclude the possibility of alternating and block

copolymerization (For details, see Supplementary Fig. 19). We deduced that the present copolymerization proceeded in a random manner.

=> To provide this information, we added a description to the main text (page 12, line 5–22) and the corresponding data to Supplementary Information (Supplementary Fig. 19).

[2-2] And do the obtained supramolecular copolymers lead to the enantio-separation of enantiomers?

=> For forming the selector supramolecular polymer to address this comment, we chose the copolymer of $1\cdot 2^S$ and $1\cdot 5^S$ because of its long-term solution stability in dodecane.

=> We demonstrated that $1\cdot 2^S$ and $1\cdot 5^S$ are copolymerizable (Supplementary Fig. 24) and that their copolymer can realize the enantio-separation of $3^S/3^R$ and $4^S/4^R$ (Supplementary Fig. 25).

=> To provide this information, we added a description to the main text (page 14, line 20–23) and the corresponding data to Supplementary Information (Supplementary Figs 24 and 25).

[3] For the enantio-separation of $3^S/3^R$, the author used two different mixing order of reagents, and got 3^S -enriched mixture with different *S*:*R* ratios and yields. The factors that are relevant to the separation efficiency and yield should be carefully discussed.

=> We conducted the enantio-separation of $3^S/3^R$ using two protocols (Supplementary Method 7, protocols 1 and 2). In protocol 1, we got the mixture of $3^S:3^R$ with 90:10 ratio and 49% yield (Fig. 6d), while in protocol 2, with 88:12 ratio and 71% yield (Supplementary Fig. 26b). Thus, the ratios were almost same, while the yields were different (protocol 2 > protocol 1).

=> In protocol 1, $1\cdot 3^S$ and $1\cdot 3^R$ are added as one portion to the dodecane phase at high concentrations. Consequently, both of them tend to form precipitate and the precipitation of $1\cdot 3^S$ reduces the yield.

=> In protocol 2, 3^S and 3^R in the solid phase are slowly dissolved to the dodecane phase and then associate with **1** to gradually form $1\cdot 3^S$ and $1\cdot 3^R$. Because such gradual feeding of $1\cdot 3^S$ can suppress its nuclei formation and its subsequent polymerization and precipitation, the yield of 3^S -enriched mixture in the supernatant becomes higher than the case of protocol 1.

=> To provide this explanation, we added a description to the main text (page 15, line 12–16).

[4-1] Does this family of helical supramolecular polymers can lead to the completely enantio-separation of relevant chiral guest molecules?

=> In the enantio-separation experiments described in the previous manuscript, only $1\cdot 2^S$ was used as a chiral selector. To prove the generality of our method, we demonstrated that other chiral selectors, such as the polymer $1\cdot 5^S$ and the copolymer of $1\cdot 2^S$ and $1\cdot 5^S$, can also realize the enantio-separation of $3^S/3^R$ and $4^S/4^R$.

=> To provide this information, we added a description to the main text (page 14, line 20–23) and the corresponding data to Supplementary Information (Supplementary Figs 23–25).

[4-2] What about the advantages over other methods of enantio-separation?

- => One unique feature of the present enantio-separation method is that the order of affinity between the selector and the guest enantiomers can be rationally predicted from the homochiral copolymerization rule, *i.e.* the selector accommodates the homo-chiral guest while rejects the hetero-chiral one.
- => To provide this information, we added a description to the main text (page 14, line 23–page 15, line 2).

Reviewers' comments:

Reviewer #1 (Remarks to the Author):

I appreciated the efforts and the detailed characterization of the copolymers in this work. But I do think that, with the inclusion of the point I raise below, the manuscript is suitable for publication in a more specialized journal.

1. My major point is that this work lacks the novelty and general interest for publication in Nature Communication. As I claimed before, this work is about the chiral molecules controlled polymers and its application in chiral guest recognition. The author highlight the "non-covalent interaction sites". However, there are a lot of examples of supramolecular assemblies which their chirality was controlled by chiral molecules. I did not accept the "enantio-separation" raised by this work. Moreover, from the aspect of enantiomer excess, I did not see the advantage of this work. Just as author stated, if the aim of this work is to prove that the concept of "monomer-connection sites" is useful for chiral recognition, I suggest to a more specialized journal.

2. about the "enantio-separation".

1) I did not overlook the supplementary method and figures. Actually, the additional experiments exactly reinforced my claim.

In Fig. 6c (Protocol 1), 1·3S (4.0 mM), 1·3R (4.0 mM), and 1·2S (4.0 mM) were mixed together, and the ratio of 3S:3R is 90:10.

While in Supplementary Fig. 26 (protocol 2, "directly separation"), 3S (4.0 mM) and 3R (4.0 mM), 1·2S (3.0 mM) and free acid 1 (1.0 mM) were mixed together, and the ratio of 3S:3R is 88:12.

Clearly, the ingredients are exactly the same, only the concentration of 2S in protocol 2 decrease from 4 mM to 3 mM. At the same time, the ratio of 3S:3R is also decreased from 90:10 to 88:12. I would say these experiments and protocols are the same, because the free acid 1 could also form 1·3S and 1·3R in the solution. Basically, target guests still need the pre-treatment in both protocols.

Moreover, as shown in Supplementary Fig. 27, without free acid 1, the ratio of 3S:3R is 67:33, the enantiomer excess (ee) is 34%. For 4S/4R, the ratio decrease to 58:42, the ee is 16%. I would accept 34%, this is the reason I said before that the selectivity of this work is not good enough. I want to emphasize that once the free acid 1 is added, it more like the chiral-recognize between two helical structures rather than the enantio-separation. In addition, up to now, this work did not separate pure 3S and 3R from the copolymer and precipitation. The ratio obtained by HPLC just shows the ability of recognition.

2) How many times did the author repeat the enantio-reorganization experiment? The statistics distribution and the error bar should be added in the paper.

3) Recycling experiments should be conducted for several reorganization runs to show the structures behaves.

3. about the double-strand helical structure

From the repose of authors, I got the conclusion that the authors still did not sure about the real morphology but chosen the double-strand helical structure to illustrate the mechanism in the whole article.

Moreover, there is no comparison between this work and the cited literature:

1) The molecular structure is different. In the work of Würthner, F. et al., the π - π stacking of perylene bisimide part is important and crucial for the self-assembly. But in this work, it more like a supramolecular systems with two or multiply components, which the assembly mainly caused by the chiral amino alcohol.

2) By the way, I suggest the author to do the experiments with chiral molecule which has only amino group or alcohol group. For example, 1-Phenylethanol, 1-Phenylethylamine or other molecules with two amino groups like 1,2-Diaminocyclohexane.

3) The author claim that the helical structures in the work of Würthner, F. et al. is a double-strand. Again, I did not find any word or description of "double-strand" in the cited paper.

4) I want to illustrate that the cartoon illustration of the helixes in the cited paper is more like a

twisted structure. The monomers aggregated mainly due to the π - π stacking and hydrogen bonding of perylene bisimide, while amide groups in two sides induced a helical packing mode. Maybe the author misunderstood this.

The last point is about the XRD (Supplementary Fig. 7)

1) Please added the details of this measurement and the method to get the crystal. For example, did the author use the same molar ratio and solvent? Otherwise it will not illustrate the problem.

2) From Supplementary Fig. 7, the crystal structures are not the helical and the complexes seem to form achiral crystals. This is different from the supramolecular polymers, which is chiral structures. Crystallization with long alkyl chains might be difficult, but I am afraid this crystal obtained with different molecular structure and experimental condition is incomparable.

In order to further illustrate the structure, I suggest

1) To do the XRD measurement of the as-formed assemblies directly (like the precipitation)

2) DFT calculation is another good option.

Reviewer #2 (Remarks to the Author):

The authors have responded carefully to the referee's comments; however, their response to comment #4 still is not convincing.

The authors claimed that homochiral combination of monomers leads to a copolymer forming homochiral helices; while, heterochiral combination does not lead to copolymerization. They have provided some AFM images of the helices formed by copolymerization of homochiral monomers, but this again cannot help to prove their claim. The referee suggests utilizing a chemical method (e.g. IR) to confirm the occurrence of copolymerization reaction in a homochiral combination (e.g. 1.2 and 1.33) by a characteristic peak, accordingly this peak should be absent in the heterochiral combination.

Reviewer #3 (Remarks to the Author):

I have read the revised manuscript and the response letter point by point. The authors have conducted all of the required revisions and carried out the additional experiments, and these revisions certainly reinforced their claims. I think that the revised manuscript now could be accepted by Nat. Commun.

Answers to Comments Raised by Editor and Reviewers

For Reviewer #3

I have read the revised manuscript and the response letter point by point. The authors have conducted all of the required revisions and carried out the additional experiments, and these revisions certainly reinforced their claims. I think that the revised manuscript now could be accepted by *Nat. Commun.*

=> We most appreciate these highly encouraging comments.

For Reviewer #2

The authors have responded carefully to the referee's comments.

=> We appreciate the highly encouraging comment, as well as the following constructive suggestion. We conducted the suggested experiment, which further reinforced our claim.

However, their response to comment [4] still is not convincing. The authors claimed that homochiral combination of monomers leads to a copolymer forming homochiral helices; while, heterochiral combination does not lead to copolymerization. They have provided some AFM images of the helices formed by copolymerization of homochiral monomers, but this again cannot help to prove their claim. The referee suggests utilizing a chemical method (*e.g.* IR) to confirm the occurrence of copolymerization reaction in a homochiral combination (*e.g.* $\mathbf{1}\cdot\mathbf{2}^S$ and $\mathbf{1}\cdot\mathbf{3}^S$) by a characteristic peak, accordingly this peak should be absent in the heterochiral combination.

=> For supporting our claim in a comprehensive manner, we took the following three approaches; two are chemical approaches (**FT-IR / CD**) and one is a physical approach (**AFM**).

=> **FT-IR:** According to the reviewer's kind suggestion, we measured the FT-IR spectra of homo-chiral ($\mathbf{1}\cdot\mathbf{2}^S$ and $\mathbf{1}\cdot\mathbf{3}^S$) and hetero-chiral ($\mathbf{1}\cdot\mathbf{2}^S$ and $\mathbf{1}\cdot\mathbf{3}^R$) mixtures in dodecane at various temperatures. When the IR spectra of the homo-chiral mixture in dodecane were measured upon cooling from 348 K to 298 K, the absorption at $1,568\text{ cm}^{-1}$, attributable to hydrogen-bonded R-CO^{2-} emerged and increased its intensity. In contrast, when the hetero-chiral mixture was measured likewise, the corresponding absorption was not observed (Supplementary Fig. 17). This result is consistent with the reviewer's suggestion and supports our claim.

To provide this information, we added a description to the main text (page 11, line 20–25) and the corresponding data to Supplementary Information (Supplementary Fig. 17).

=> **CD:** As a widely used method for distinguishing whether two kinds of monomers are co-assembled to form a copolymer or self-sorted into their homopolymers (see *refs.* 44, 48 and 51), we conducted the CD study on monomer mixtures, as follows.

As described in our previous manuscript, we already conducted the CD study on the equimolar mixtures, which supported our claim. Thus, the CD spectrum of the homo-chiral mixture ($\mathbf{1}\cdot\mathbf{2}^S:\mathbf{1}\cdot\mathbf{3}^S = 50:50$) was largely different from the average of the spectra of unmixed monomers, indicating that they co-assembled to form a copolymer (Fig. 5a). Meanwhile, the CD spectrum of the hetero-chiral mixture ($\mathbf{1}\cdot\mathbf{2}^S:\mathbf{1}\cdot\mathbf{3}^R = 50:50$) was very close to the average of the spectra of unmixed monomers, indicating that they self-sorted into two kinds of homopolymers (Fig. 5b).

For further validation of our claim, we newly conducted the same CD study on the mixtures with various ratios of the monomers, for both of the homo-chiral ($\mathbf{1}\cdot\mathbf{2}^S:\mathbf{1}\cdot\mathbf{3}^S = 90:10$ to $10:90$; Supplementary Fig. 15) and hetero-chiral ($\mathbf{1}\cdot\mathbf{2}^S:\mathbf{1}\cdot\mathbf{3}^R = 90:10$ to $10:90$; Supplementary Fig. 16) combinations. The results were again consistent with our claim.

To provide the information of the new experiment, we added a description to the main text (page 11, line 15–20) and the corresponding data to Supplementary Information (Supplementary Figs 15 and 16).

=> **AFM:** According to the above results, the hetero-chiral combination afforded two kinds of homopolymers. Their morphologies were already known to be fibrous helices, as characterized for the unmixed monomers. Meanwhile, the co-assembly formed in the homo-chiral combination was a new species, whose morphology could not be determined by the FT-IR and CD measurements. Therefore, we measured AMF images of the homo-chiral mixtures with various ratios of the monomers.

Irrespective of the ratio of the monomers, a lot of fibrous helices with a uniform helical pitch were observed in the AFM images. Furthermore, upon gradually changing the ratio of the two monomers, the helical pitch of the fibres also gradually changed (Supplementary Fig. 23). Therefore, we concluded that (i) the morphology of the homo-chiral co-assembly was a fibrous helix and that (ii) the two homo-chiral monomers were copolymerized almost randomly so that the pitch of the helix was uniform and determined by the ratio of the monomers.

Before this revision, this information has already been provided in the main text (page 12, line 24–page 13, line 16) and Supplementary Information (Supplementary Fig. 23).

For Reviewer #1

I appreciated the efforts and the detailed characterization of the copolymers in this work. But I do think that, with the inclusion of the point I raise below, the manuscript is suitable for publication in a more specialized journal.

- => We appreciate many suggestions by Reviewer #1. We fully addressed comments [2-2], [2-3], [2-4], [3], and [4], with conducting additional experiments. We believe that Reviewer #1 would now admit our answers to these comments. Meanwhile, comment [1] is a summary of comment [2]. Therefore, only a remaining issue would be comment [2-1].
- => The point of comment [2-1] is very simple; whether our demonstration in this work can be regarded as ‘enantio-separation’ or not. We should state that the claim of Reviewer #1 in comment [2-1] is scientifically incorrect, as described in our answer. Our confidence is supported by the comments of Reviewers #2 and #3. We are also requesting the editor to have a third-party variation. We believe that Reviewer #1 would kindly understand this time.
- => For fair review, we would like to ask Reviewer #1 to keep in mind our requests as follows.
 - If you would still claim that what we demonstrated cannot be regarded as ‘enantio-separation’, please clearly state your definition of ‘enantio-separation’, hopefully with referring literature that supports the validity of your definition.
 - If you would still claim the lack of novelty of this work, please specifically refer papers that report the separation of a racemic mixture into an enantio-enriched fraction by using a solution-phase supramolecular polymer as a chiral selector.
 - If you would further request additional structural characterization of our supramolecular polymer, please refer several papers of solution-phase supramolecular polymers, for proving that our structural analysis falls short of the standard level of this field.

[1] Major point

[1-1] My major point is that this work lacks the novelty and general interest for publication in *Nature Communication*. As I claimed before, this work is about the chiral molecules controlled polymers and its application in chiral guest recognition. The author highlight the “non-covalent interaction sites”. However, there are a lot of examples of supramolecular assemblies which their chirality was controlled by chiral molecules.

=> This work reports the first demonstration that a racemic mixture can be separated into enantio-enriched fractions through the interaction with a solution-phase supramolecular polymer. If the reviewer does not agree with it, he/she should specifically refer similar previous examples.

[1-2] I did not accept the “enantio-separation” raised by this work. Moreover, from the aspect of enantiomer excess, I did not see the advantage of this work. Just as author stated, if the aim of this work is to prove that the concept of “monomer-connection sites” is useful for chiral recognition, I suggest to a more specialized journal.

=> For the definition of ‘enantio-separation’, please see our answer to comments [2-1]. For typical enantiomer excesses of other chiral selectors, please see our answer to comment [2-2].

=> This work demonstrated that the monomer-connection sites of a supramolecular polymer are useful for enantio-separation of a racemic guest. Among all molecular recognition events, chiral recognition is one of the most difficult ones. Therefore, our achievement strongly implies various potential utilities of supramolecular polymers.

[2] About the “enantio-separation”

[2-1] I did not overlook the supplementary method and figures. Actually, the additional experiments exactly reinforced my claim. In Fig. 6c (protocol 1), $1\cdot 3^S$ (4.0 mM), $1\cdot 3^R$ (4.0 mM), and $1\cdot 2^S$ (4.0 mM) were mixed together, and the ratio of $3^S:3^R$ is 90:10. While in Supplementary Fig. 26 (protocol 2, “directly separation”), 3^S (4.0 mM) and 3^R (4.0 mM), $1\cdot 2^S$ (3.0 mM) and free acid **1** (1.0 mM) were mixed together, and the ratio of $3^S:3^R$ is 88:12. Clearly, the ingredients are exactly the same, only the concentration of 2^S in protocol 2 decrease from 4 mM to 3 mM. At the same time, the ratio of $3^S:3^R$ is also decreased from 90:10 to 88:12.

=> First of all, there is a large difference between protocol 1 and protocol 2 in practical aspects. In protocol 1, the racemic mixture was prepared by mixing enantiopure $1\cdot 3^S$ and $1\cdot 3^R$. Meanwhile, in protocol 2, the starting material is a racemic mixture of 3^S and 3^R .

I would say these experiments and protocols are the same, because the free acid **1** could also form $1\cdot 3^S$ and $1\cdot 3^R$ in the solution. Basically, target guests still need the pre-treatment in both protocols.

=> To clarify the point, let us summarize the reviewer’s claim in comment [2-1]. The reviewer compares the following two cases of enantio-enrichment.

Case (i): enantio-enrichment caused by the enantioselective exchange of the guest molecules (free from **1**) between the inside and outside of the host supramolecular polymer.

Case (ii): enantio-enrichment caused by the enantioselective insertion of the salt pairs (**1**–guest molecule) into the host supramolecular polymer.

We proved that the enantio-enrichment in this work is caused by the mechanism of case (ii). The reviewer claims that case (ii) is not ‘enantio-separation’ and that this work is not novel.

=> The point is whether case (ii) can be regarded as ‘enantio-separation’ or not. We should state that the reviewer’s above claim is scientifically incorrect, because:

In terms of terminology, both of case (i) and case (ii) are regarded as ‘enantio-separation’; the starting material is racemic, and the product is enantio-enriched.

In terms of novelty, both of case (i) and case (ii) have never been reported for solution-phase supramolecular polymers.

In terms of mechanism, both of case (i) and case (ii) utilize the monomer-connection sites of the supramolecular polymer for differentiating the enantiomers of the guest molecules.

In terms of operation, both of case (i) and case (ii) do not require chemical reaction for the attachment and detachment of **1** to the guest molecules.

=> If the reviewer does not accept our statement as above, he/she should clarify his/her definition of ‘enantio-separation’ and logically explain why case (ii) does not meet the definition, hopefully with citing literature that proves the validity of his/her definition.

Moreover, as shown in Supplementary Fig. 32, without free acid **1**, the ratio of $3^S:3^R$ is 67:33, the enantiomer excess is 34%. For $4^S/4^R$, the ratio decrease to 58:42, the ee is 16%. I would accept 34%, this is the reason I said before that the selectivity of this work is not good enough.

=> Supplementary Fig. 32 shows negative control experiments for confirming the role of free acid **1**. The enantioselectivity of these experiments does not affect the importance of this work.

I want to emphasize that once the free acid **1** is added, it more like the chiral-recognize between two helical structures rather than the enantio-separation.

=> In this comment, the reviewer should clarify the definition of ‘chiral recognition’ and ‘enantio-separation’, and logically explain why case (ii) does not meet the definition of ‘enantio-separation’ hopefully with citing literature that proves the validity of his/her definition.

[2-2] In addition, up to now, this work did not separate pure **3^S** and **3^R** from the copolymer and precipitation. The ratio obtained by HPLC just shows the ability of recognition.

=> The term ‘enantio-separation’ is generally used for processes that afford enantio-enriched but not enantio-pure materials from racemic mixtures. In the following studies on supramolecular chiral selectors, achieved selectivities were comparable or inferior to that this work, and the term ‘enantio-separation’ was used.

Edwards, W. *et al. J. Am. Chem. Soc.* **136**, 1116 (2014).

Wu, K. *et al. Nat. Commun.* **5**, 10487 (2016).

Pinxterhuis, E. B. *et al. Chem. Sci.* **8**, 6409 (2017).

Ousaka, N. *et al. Nat. Commun.* **10**, 1457 (2019).

Huang, Y. *et al. Angew. Chem. Int. Ed.* **58**, 10859 (2019).

[2-3] How many times did the author repeat the enantio-reorganization experiment? The statistics distribution and the error bar should be added in the paper.

=> According to our literature survey, the suggested statistical analysis has generally not been done in the studies on enantio-separation experiments, unlike some biological tests. For example, please see the references in our answer to comment [2-2].

=> Just in case, we repeated the enantio-separation of **3^S/3^R** and **4^S/4^R**, two more times for each, and found that our enantio-separation could be performed in good reproducibility, in terms of both yield and selectivity (Supplementary Fig. 26a,b).

=> To provide this information, we added a description to the main text (page 15, line 7–8) and the corresponding data to Supplementary Information (Supplementary Fig. 26a,b).

[2-4] Recycling experiments should be conducted for several reorganization runs to show the structures behaves.

=> In response to this suggestion, we conducted the recycling experiment. Thus, after the enantio-separation process of Fig. 6c, the carboxylic acid **1** was recovered from the supernatant and precipitate and reused in the second cycle of the enantio-separation (Supplementary Method 11), where the yield and selectivity were essentially identical to those of the first cycle (Supplementary Fig. 26c).

=> To provide this information, we added a description to the main text (page 15, line 8–11) and the corresponding method (Supplementary Method 11) and data (Supplementary Fig. 26c) to Supplementary Information.

[3] About the double-strand helical structure

=> Before answering to [3] and [4], we would like to claim that the level of structural characterization in this work is comparable or even superior to most works on solution-phase supramolecular polymers, even though the core part of this work is not structure but function. Please also note that the other two reviewers (Reviewer #2 and Reviewer #3) did not raise any concern about the structural characterization of our supramolecular polymer. If the reviewer would request further structural characterization, he/she should refer several reports on solution-phase supramolecular polymers, for proving that our structural characterization is really insufficient.

[3-1] From the repose of authors, I got the conclusion that the authors still did not sure about the real morphology but chosen the double-strand helical structure to illustrate the mechanism in the whole article.

= We previously reported a supramolecular polymer in a liquid crystalline state, which was also composed of a carboxylic acid and a chiral amino alcohol. Based on the determination of the space group ($P6_122$) by synchrotron XRD studies, we obtained precise positions of molecular components in this supramolecular polymer, which adopted a double-helical structure (*Nat. Commun.* **6**, 8418 (2015)). Please note that the space group of $P6_122$ intrinsically possesses a double-helical structure. Thus, this space group contains a right-handed six-fold screw axis. This space group also has a two-fold screw axis that is vertical to the six-fold one. Therefore, two six-fold helices become antiparallel with one another to form a double-helical structure.

=> Taking advantage of this experience, we also characterized the supramolecular polymer of **1·2^S** in a liquid crystalline state by synchrotron XRD measurement (Supplementary Fig. 8). The XRD pattern was quite resembling to that in our previous report (*Nat. Commun.* **6**, 8418 (2015)) and could be well elucidated with supposing the space group of $P6_122$. Thus, we proposed a double-helical structure as a possible model of the supramolecular polymer of **1·2^S**.

=> We had the XRD data before the first submission and had originally planned to publish them in a separate paper. However, for properly answering the comment by the reviewer, we decided to provide the above information by adding a description to the main text (page 7, line 10–19) and the corresponding method and data to Supplementary Information (Supplementary Method 6 and Supplementary Fig. 8).

[3-2] Moreover, there is no comparison between this work and the cited literature: The molecular structure is different. In the work of Würthner, F. *et al.*, the π - π stacking of perylene bisimide part is important and crucial for the self-assembly. But in this work, it more like a supramolecular systems with two or multiply components, which the assembly mainly caused by the chiral amino alcohol.

=> Before answering this comment, we would like to emphasize that this debate, *i.e.* the classification and comparison of our helical assembly and that of Würthner, F. *et al* (comments [3-2], [3-4], and [3-5]) is not productive, because it is hardly related to the value of this work. Therefore, we would at first briefly explain the morphological characteristics of our helical polymer and that of Würthner, F. *et al* in a separate manner. Then, we would also provide the answer to the previous question of the reviewer, which is the origin of this debate.

=> As described in our answer to comment [3-1], XRD studies based on the determination of space group suggested that the helical assembly of the present work adopts a resembling structure to that of the helical assembly in our previous work (*Nat. Commun.* **6**, 8418 (2015)). In a monomeric unit, two salt pairs are arranged in a symmetric relationship (see right figure). In its helical stack, bulky carboxylic-acid molecules are arranged in two independent helical arrays. In morphology observation measurements such as AFM, these two helices are considered to be preferentially visualized because of their bulkiness.

=> In the helical assembly of Würthner, F. *et al.* (*J. Am. Chem. Soc.* **137**, 3300 (2015)), the monomeric unit adopts a dumbbell shape with a core and two bulky peripheries (see right figure). In its helical stack, we can find two independent helical arrays of the bulky peripheries. In morphology observation measurements such as AFM, these two helices are considered to be preferentially visualized because of their bulkiness.

In relation to this, Percec, V. *et al.* reported helical assemblies of dumbbell-shape perylene bisimides, whose molecular structures are very similar to that of Würthner, F. *et al.* These assemblies are denoted as ‘double helices’ in the reports of Percec, V. *et al.* (*Nat. Chem.* **8**, 80 (2016); *J. Am. Chem. Soc.* **141**, 15761 (2019)).

=> Let us remind the reviewer the origin of this debate. In the first round of review, the reviewer commented that the fibres in the AFM images of this work looked like single helices, although the proposed structure was double-helical. In our proposed structure, two helices with the same shape are aligned along the same axis. When we observe the whole structure from the direction vertical to the axis, the turns of the two helices appear alternatingly toward our side. Please see the following schematic representation, where two coaxial helices are equivalent but depicted in different colours for clarity.

=> Several nanostructures composed of helically arrayed dumbbell-shaped units have been reported, including those composed of single-component and multi-component dumbbells. Some of them provide AFM images similar to the above, in which the pitches of two co-axial helices appear alternatingly.

- Iwaura, R. *et al.* *J. Am. Chem. Soc.* **128**, 13298 (2006).
- Stepanenko, V. *et al.* *Chem. Eur. J.* **19**, 4176 (2013).
- Ogi, S. *et al.* *J. Am. Chem. Soc.* **137**, 3300 (2015).

[3-3] By the way, I suggest the author to do the experiments with chiral molecule which has only amino group or alcohol group. For example, 1-phenylethanol, 1-phenylethylamine or other molecules with two amino groups like 1,2-diaminocyclohexane.

=> We appreciate the reviewer's kind suggestion, but we did not use the suggested molecules this time, because of the following reason.

=> We have already confirmed that two analogues of 2^S , one of which lacks an alcohol group ($2_{(OMe)}^S$) and the other lacks a primary amine group ($2_{(NMe)}^S$) cannot form supramolecular polymers (Supplementary Fig. 3), indicating the crucial role of hydrogen-bonding interactions.

=> Compared with the suggested molecules, the molecular structures of the above analogues ($2_{(OMe)}^S$ and $2_{(NMe)}^S$) are closer to that 2^S . Therefore, for the purpose of specifying which structural elements are essential for the supramolecular polymerization, the above analogues ($2_{(OMe)}^S$ and $2_{(NMe)}^S$) are more critical than the suggested molecules.

[3-4] The author claim that the helical structures in the work of Würthner, F. *et al.* is a double-strand. Again, I did not find any word or description of "double-strand" in the cited paper.

=> Please see our answer to comment [3-2].

[3-5] I want to illustrate that the cartoon illustration of the helices in the cited paper is more like a twisted structure. The monomers aggregated mainly due to the π - π stacking and hydrogen bonding of perylene bisimide, while amide groups in two sides induced a helical packing mode. Maybe the author misunderstood this.

=> Please see our answer to comment [3-2].

- [4] The last point is about the XRD (Supplementary Fig. 7):
- [4-1] Please added the details of this measurement and the method to get the crystal. For example, did the author use the same molar ratio and solvent? Otherwise it will not illustrate the problem.
- => These crystal structures were obtained not by our own measurement but from our comprehensive survey of Cambridge Structural Database for deducing the pattern of hydrogen-bond network. No less than 16 kinds of the salts of carboxylic acids and **2^S** (including derivatives) were found to adopt the same hydrogen-bond network shown in Fig. 2h. As the representatives, two examples (CCDC188/130 and CCDC 619698) were shown in Supplementary Fig. 7. Thus, our proposed network is supported with statistic survey.
- => These crystals were formed by mixing a carboxylic acid and an amino alcohol in 1:1 molar ratio, similar to the case of our supramolecular polymer. These crystals were obtained by the recrystallization from alcohol or water, while our supramolecular polymer was formed in dodecane. Despite the inconsistency of solvents, the hydrogen-bond network in the crystal structure may have common characteristics to the network in the supramolecular polymer, because in both cases, solvent molecules are not involved in the hydrogen-bond network of the carboxylic acid and the amino alcohol.
- => The above discussion has been generally accepted and widely used. There have been a number of reports in which hydrogen-bond networks of supramolecular polymers were deduced from the crystal structures of their non-alkyl-chain analogues, with admitting the inconsistency of solvents for supramolecular polymerization and for recrystallization.
- Lightfoot, M. P. *et al. Chem. Commun.* 1945 (1999).
Ishida, Y. *et al. J. Am. Chem. Soc.* **124**, 14017 (2002).
Ky Hirschberg, J. H. K. *et al. Chem. Eur. J.* **9**, 4222 (2003).
Nakamura, K. *et al. Chem. Commun.* **52**, 7157 (2016).
- [4-2] From Supplementary Fig. 7, the crystal structures are not the helical and the complexes seem to form achiral crystals. This is different from the supramolecular polymers, which is chiral structures. Crystallization with long alkyl chains might be difficult, but I am afraid this crystal obtained with different molecular structure and experimental condition is incomparable.
- => Please note that these crystal structures have a helical chirality, as can be seen in their space groups ($P2_12_12_1$ and $P2_1$). Their helical chirality is apparently not obvious because they are two-fold screws. To provide this information, we added the space groups and the direction of the 2_1 -helical axes of these crystal structures to **Supplementary Fig. 7**.
- => The twisting mode of these crystal structures is two-fold screw, while that of the supramolecular polymer should be different due to the steric demand of long alkyl chains. This is the reason why we used the information of crystal structures only for deducing the pattern of hydrogen-bond network.
- => For deducing the whole structure, we combined the information from CD, AFM, and XRD measurements. This is a very common approach in the studies of solution-phase supramolecular polymers.
- [4-3] In order to further illustrate the structure, I suggest to do the XRD measurement of the as-formed assemblies directly (like the precipitation). DFT calculation is another good option.
- => Please see our answer to comment [3-1].

REVIEWERS' COMMENTS:

Reviewer #2 (Remarks to the Author):

[Reviewer #2 looked over the author's response to reviewer #1's comments as well.] The authors have clearly addressed and responded to the referee's last concern and provided decent experimental evidence. This referee recommends publishing the revised manuscript.